# Cause-Effect Inference in Location-Scale Noise Models: Maximum Likelihood vs. Independence Testing

**Xiangyu Sun**
Simon Fraser University
`xiangyu_sun@sfu.ca`

**Oliver Schulte**
Simon Fraser University
`oschulte@cs.sfu.ca`

## Abstract

A fundamental problem of causal discovery is cause-effect inference, learning the correct causal direction between two random variables. Significant progress has been made through modelling the effect as a function of its cause and a noise term, which allows us to leverage assumptions about the generating function class. The recently introduced heteroscedastic location-scale noise functional models (LSNMs) combine expressive power with identifiability guarantees. LSNM model selection based on maximizing likelihood achieves state-of-the-art accuracy, when the noise distributions are correctly specified. However, through an extensive empirical evaluation, we demonstrate that the accuracy deteriorates sharply when the form of the noise distribution is misspecified by the user. Our analysis shows that the failure occurs mainly when the conditional variance in the anti-causal direction is smaller than that in the causal direction. As an alternative, we find that causal model selection through residual independence testing is much more robust to noise misspecification and misleading conditional variance.

## 1 Introduction

Distinguishing cause and effect is a fundamental problem in many disciplines, such as biology, healthcare and finance [31, 8, 13]. Randomized controlled trials (RCTs) are the gold standard for finding causal relationships. However, it may be unethical, expensive or even impossible to perform RCTs in real-world domains [20, 18]. Causal discovery algorithms aim to find causal relationships given observational data alone. Traditional causal discovery algorithms can only identify causal relationships up to Markov equivalence classes (MECs) [26, 10, 3]. To break the symmetry in a MEC, additional assumptions are needed [20, 22], such as the type of functional dependence of effect on cause. Structural causal models (SCMs) specify a functional class for the causal relations in the data [17, 20]. In this work, we focus on one particular type of SCM called location-scale noise models (LSNMs) [29, 11, 30, 28, 9]:

$$Y := f(X) + g(X) \cdot Z_Y \tag{1}$$

where $X$ is the cause, $Y$ is the effect, written $X \rightarrow Y$, and $Z_Y$ is a latent noise variable independent of $X$ (i.e., $X \perp\!\!\!\perp Z_Y$). The functions $f$ and $g$ are twice-differentiable on the domain of $X$, and $g$ is strictly positive on the domain of $X$. LSNMs generalize the widely studied additive noise models (ANMs), where $g(x) = 1$ for all $x$. ANMs assume a constant conditional variance $\mathbb{V}[Y|X]$ for different values of $X$, which can limit their usefulness when dealing with heteroscedastic noise in real-world data. LSNMs allow for heteroscedasticity, meaning that $\mathbb{V}[Y|X]$ can vary depending on the value of $X$. In our experiments on real-world data, methods assuming an LSNM outperform those assuming ANM.

37th Conference on Neural Information Processing Systems (NeurIPS 2023).

We follow previous work on cause-effect inference and focus on the bivariate setting [29, 11, 30, 9]. There are known methods for leveraging bivariate cause-effect inference in the multivariate setting (see Appendix A). Two major methods for selecting a causal SCM direction in cause-effect inference are maximum likelihood (ML) and independence testing (IT) of residuals vs. the (putative) cause [14]. Both have been recently shown good accuracy with LSNMs [11, 9], especially when the $f$ and $g$ functions are estimated by neural networks. Immer et al. [9] note, however, that ML can be less robust than IT, in the sense that accuracy deteriorates when the noise distribution is not Gaussian.

In this paper we investigate the robustness of ML vs. IT for LSNMs. Our analysis shows that ML cause-effect inference performs poorly when two factors coincide: (1) *Noise Misspecification*: the ML method assumes a different form of the noise distribution from the true one. (2) *Misleading Conditional Variances (CVs)*: $\mathbb{V}[Y|X] > \mathbb{V}[X|Y]$ in the data generated by causal direction $X \to Y$. For example, in the experiment on synthetic datasets shown in Table 1 below, (i) changing the true noise distribution from Gaussian to uniform and (ii) manipulating $\mathbb{V}[Y|X]$, while keeping other settings equal, can decrease the rate of identifying the true causal direction from 100% to 10%. In contrast, IT methods maintain a perfect 100% accuracy. The difference occurs with and without data standardization, and with increasing sample sizes.

Both conditions (1) and (2) often hold in practice. For real-world domains, assumptions about the noise distribution can be hard to determine or verify. It is also common to have misleading CVs in real-world datasets. For example, in the Tübingen Cause-Effect Pairs benchmark [14], about 40% of the real-world datasets exhibit a misleading CV (see Table 7 in the appendix).

We make the following contributions to understanding location-scale noise models (LSNMs):

- Describe experiments and theoretical analysis to show that ML methods succeed when the form of the noise distribution is known. In particular, ML methods are then robust to incorrectly specifying a noise variance parameter.
- Demonstrate empirically that ML methods often fail when the form of the noise distribution is misspecified and CV is misleading, and analyze why.
- Introduce a new IT method based on an affine flow LSNM model.
- Demonstrate, both theoretically and empirically, that our IT method is robust to noise misspecification and misleading CVs.

The paper is structured as follows. We discuss related works and preliminaries in Section 2 and Section 3, respectively. Section 4 examines when and why ML methods fail. Section 5 demonstrates the robustness of the IT method. Experiments with 580 synthetic and 99 real-world datasets are given in Section 7. The code and scripts to reproduce all the results are given online [1].

## 2   Related works

*Causal Discovery.* Causal discovery methods have been widely studied in machine learning [26, 10, 3]. Assuming causal sufficiency and faithfulness, they find causal structure up to a MEC. To identify causal relations within a MEC, additional assumptions are needed [20, 22]. SCMs exploit constraints that result from assumptions about the functional dependency of effects on causes. Functional dependencies are often studied in the fundamental case of two variables $X$ and $Y$, the simplest MEC. Mooij et al. [14] provide an extensive overview and evaluation of different cause-effect methods in the ANM context. We follow this line of work for LSNMs with possibly misspecified models.

*Structural Causal Models.* Assuming a linear non-Gaussian acyclic model (LiNGAM) [24, 25], the causal direction was proved to be identifiable. The key idea in DirectLiNGAM [25] is that in the true causal direction $X \to Y$, the model residuals are independent of $X$. We refer to methods derived from the DirectLiNGAM approach as *independence testing* (IT) methods. The more general ANMs [7, 19] allow for nonlinear cause-effect relationships and are generally identifiable, except for some special cases. There are other identifiable SCMs, such as causal additive models (CAM) [2], post-nonlinear models [32] and Poisson generalized linear models [16].

*LSNM Identifiability.* There are several identifiability results for the causal direction in LSNMs. Xu et al. [30] prove identifiability for LSNMs with linear causal dependencies. Khemakhem et al. [11]

---

[1]https://github.com/xiangyu-sun-789/CAREFL-H

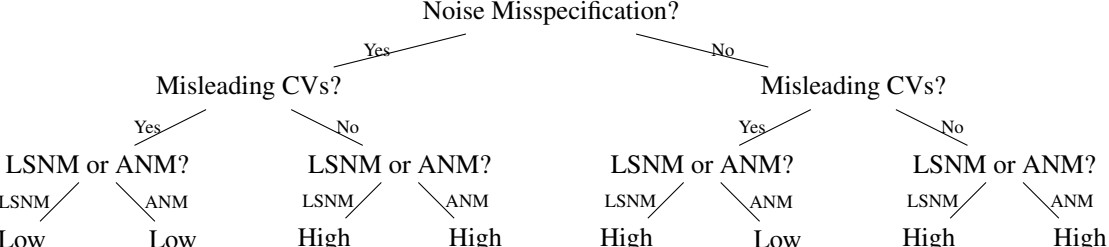

Figure 1: Summary of the accuracy results for ML methods for LSNMs and ANMs with respect to noise misspecification and misleading CVs.

show that nonlinear LSNMs are identifiable with Gaussian noise. Strobl and Lasko [28], Immer et al. [9] prove LSNMs are identifiable except in some pathological cases (see Appendix C).

*Cause-Effect Inference in LSNMs.* The generic model selection blueprint is to fit two LSNM models for each direction $X \rightarrow Y$ and $X \leftarrow Y$ and select the direction with a higher model score. HECI [30] bins putative cause values and selects a direction based on the Bayesian information criterion. BQCD [29] uses nonparametric quantile regression to approximate minimum description length to select the direction. GRCI [28] finds the direction based on mutual information between the putative cause and the model residuals. LOCI [9] models the conditional distribution of effect given cause with Gaussian natural parameters. Then, it chooses the direction based on either likelihood (LOCI-M) or independence (LOCI-H). CAREFL-M [11] fits affine flow models and scores them by likelihood. CAREFL-M is more general than LOCI since LOCI uses a fixed Gaussian prior, whereas CAREFL-M can utilize different prior distributions. DECI [4] generalizes CAREFL-M to multivariate cases for ANMs. Unlike our work, none of these papers provide a theoretical analysis of noise misspecification and misleading CVs in LSNMs.

*Noise Misspecification.* Schultheiss and Bühlmann [23] present a theoretical analysis of model misspecification pitfalls when the data are generated by an ANM. Their results focus on model selection with Gaussian likelihood scores, which use Gaussian noise distributions. They suggest that IT-based methods may be more robust to model misspecification in ANMs. The authors also briefly discuss the complication of fitting data generated by a ground-truth ANM model with an LSNM model. Since the backward model of an ANM often takes the form of an LSNM, fitting data generated by a ground-truth ANM with an LSNM model may result in an incorrect causal direction. While their analysis examines ANM data, the ground-truth model in our work is an LSNM. The $g()$ scale term in Equation (1), which is absent in ANMs, plays a critical role in LSNMs.

*Conditional Variance and Causality.* To the best of our knowledge, the problem of misleading CVs has not been identified nor analyzed in previous work for LSNMs. Prior studies that have focused on CVs were based on ANMs, and stated assumptions about how CVs relate to causal structure, to be leveraged in cause-effect inference [1, 15]. We do not make assumptions about CVs, but analyze them to understand why noise misspecification impairs cause-effect inference in LSNMs.

## 3 Preliminaries

In this section, we define the cause-effect inference problem and review the LSNM data likelihood.

### 3.1 Problem definition: cause-effect inference

Cause-effect inference takes as input a dataset $D$ over two random variables $X, Y$ with $N$ observational data pairs $(X, Y) = \{(x_1, y_1), (x_2, y_2), \ldots, (x_N, y_N)\}$ generated by a ground-truth LSNM in Equation (1). The binary output decision indicates whether $X$ causes $Y$ ($X \rightarrow Y$) or $Y$ causes $X$ ($X \leftarrow Y$). We assume no latent confounders, selection bias or feedback cycles ([14, 29, 11, 9, 30]).

Table 1: **Accuracy** over 10 datasets generated by SCM **LSNM-sine-tanh** (definition in Appendix M) with $N = 10,000$ samples. The task is a binary decision whether $X$ causes $Y$ or $Y$ causes $X$. $X$ denotes the ground-truth cause and $Y$ denotes the ground-truth effect. We rewrite Equation (1) as $Y = f(X) + \alpha \cdot g(X) \cdot Z_Y$, where $\alpha$ is a scale factor to alter the CVs. CVs are computed by binning the putative cause. We used $Gaussian(0,1)$ as model prior for both CAREFL and LOCI. The suffix *-M* denotes a ML method. The suffix *-H* denotes the corresponding IT method (see Section 5). All the datasets are standardized to have mean 0 and variance 1.

| True Noise | $Gaussian(0,1)$ | | | | | $Uniform(-1,1)$ | | | | |
|---|---|---|---|---|---|---|---|---|---|---|
| $\alpha$ | 0.1 | 0.5 | 1 | 5 | 10 | 0.1 | 0.5 | 1 | 5 | 10 |
| $\overline{\mathbb{V}}[Y\|X]$ | 0.166 | 0.615 | 0.834 | 0.990 | 0.997 | 0.044 | 0.404 | 0.673 | 0.975 | 0.994 |
| $\overline{\mathbb{V}}[X\|Y]$ | 0.455 | 0.709 | 0.793 | 0.821 | 0.817 | 0.047 | 0.375 | 0.566 | 0.681 | 0.677 |
| Percentage of Datasets With Misleading CVs | 30% | 50% | 70% | 100% | 100% | 30% | 90% | 100% | 100% | 100% |
| CAREFL-M | 1.0 | 1.0 | 1.0 | 1.0 | 1.0 | 0.7 | 0.6 | 0.7 | 0.1 | 0.1 |
| LOCI-M | 1.0 | 1.0 | 1.0 | 1.0 | 1.0 | 0.7 | 0.6 | 0.7 | 0.1 | 0.1 |
| CAREFL-H | 1.0 | 1.0 | 1.0 | 1.0 | 1.0 | 0.9 | 1.0 | 1.0 | 1.0 | 1.0 |
| LOCI-H | 1.0 | 1.0 | 1.0 | 1.0 | 1.0 | 0.7 | 1.0 | 1.0 | 1.0 | 1.0 |

## 3.2 Definition of maximum likelihood for LSNMs

An LSNM model for two variables $(X, Y)$ is a pair $(\rightarrow, P_Z)$. The $\rightarrow$ represents the direction $X \rightarrow Y$ with parameter space $f \equiv f_\theta, g \equiv g_\psi, P_X \equiv P_{X,\zeta}$. The $\leftarrow$ represents the direction $X \leftarrow Y$ with parameter space $h \equiv h_{\theta'}, k \equiv k_{\psi'}, P_Y \equiv P_{Y,\zeta'}$. For notational simplicity, we treat the model functions directly as model parameters and omit reference to their parameterizations. For example, we write $f \equiv f_\theta$ for a function $f$ that is implemented by a neural network with weights $\theta$. We refer to $P_Z$ as the model prior noise distribution.

A parameterized LSNM model defines a data distribution as follows [9] (Appendix B):

$$
\begin{aligned}
P_{\rightarrow,P_Z}(X,Y;f,g,P_X) &= P_X(X) \cdot P_{Z_Y}\Big(\frac{Y - f(X)}{g(X)}\Big) \cdot \frac{1}{g(X)} \\
P_{\leftarrow,P_Z}(X,Y;h,k,P_Y) &= P_Y(Y) \cdot P_{Z_X}\Big(\frac{X - h(Y)}{k(Y)}\Big) \cdot \frac{1}{k(Y)}
\end{aligned}
\tag{2}
$$

For conciseness, we use shorter notation $P_{\rightarrow,P_Z}(X,Y)$ and $P_{\leftarrow,P_Z}(X,Y)$ in later sections. The ML method computes the ML parameter estimates and uses them to score the $\rightarrow$ (or $\leftarrow$) model:

$$
\widehat{f}, \widehat{g}, \widehat{P_X} := \arg \max_{f,g,P_X} \prod_{i=1}^{N} P_{\rightarrow,P_Z}(x_i, y_i; f, g, P_X)
\tag{3}
$$

$$
L_{\rightarrow,P_Z}(D) := \prod_{i=1}^{N} P_{\rightarrow,P_Z}(x_i, y_i; \widehat{f}, \widehat{g}, \widehat{P_X})
\tag{4}
$$

## 4 Non-identifiability of LSNMs with noise misspecification

In this section, we show that ML methods fail under noise misspecification and misleading CVs, and analyse why. Figure 1 summarizes the diverse settings considered below.

Existing identifiability results for LSNM ML methods (Appendix C) require knowing the ground-truth noise distribution. We conducted a simple experiment to show that with noise misspecification, ML model selection can fail badly even on large sample sizes. Table 1 shows results for CAREFL-M [11] and LOCI-M [9] with model prior distribution $Gaussian(0,1)$. For the left half of the table, the data

was generated with correctly specified $Gaussian(0,1)$ noise. In this case, both CAREFL-M and LOCI-M work well, and increasing CV in the causal direction does not affect their accuracy. For the right half of the table, the data was generated with misspecified $uniform(-1,1)$ noise. With noise misspecification, the accuracy of both CAREFL-M and LOCI-M decreases to $70\%$. With both noise misspecification and misleading CVs, their accuracy becomes even lower. For example, in the last column when $\overline{\mathbb{V}}[Y|X] = 0.994$ and $\overline{\mathbb{V}}[X|Y] = 0.677$, they give an accuracy of just $10\%$.

The following results help explain why ML methods fail under noise misspecification and misleading CVs.

**Lemma 4.1.** *For an LSNM model $Y := f(X) + g(X) \cdot Z_Y$ where $Z_Y$ is the noise with mean $\mu$ and variance $\sigma^2 \neq 0$, there exists a likelihood-equivalent LSNM Model $Y := f'(X) + g'(X) \cdot Z'_Y$ such that $Z'_Y = \frac{Z_Y - \mu}{\sigma}$, $P_{\to,P_Z}(Y|X; f, g) = P_{\to,P_{Z'}}(Y|X; f', g')$, and $\mathbb{V}_{Z_Y}[Y|X] = \mathbb{V}_{Z'_Y}[Y|X]$.*

The proof is in Appendix E. Lemma 4.1 shows that for any (non-deterministic) LSNM model there is an equivalent standardized LSNM model with $\mathbb{V}[Z_Y] = 1$.

**Theorem 4.2.** $\frac{\partial \log P_{\to,P_Z}(Y|X)}{\partial \mathbb{V}[Y|X]} < 0$ *for LSNM models.*

The proof is in Appendix F. Theorem 4.2 shows that the model likelihood $P_{\to,P_Z}(Y|X)$ and the conditional variance $\mathbb{V}[Y|X]$ are negatively related (similarly for $P_{\leftarrow,P_Z}(X|Y)$ and $\mathbb{V}[X|Y]$).

**Corollary 4.3.** *Let $\to$ denote the causal model and $\leftarrow$ denote the anti-causal model. When $\mathbb{V}[Y|X]$ increases and $\mathbb{V}[X|Y]$ decreases in the data, the log-likelihood model difference $\log P_{\to,P_Z}(X,Y) - \log P_{\leftarrow,P_Z}(X,Y)$ decreases.*

The proof is in Appendix G. Under the identifiability assumption [11, 9] that ML is consistent under correct model specification, Lemma 4.1 also implies that ML remains consistent under misspecifying the variance of the noise distribution, as long as the form of the prior distribution matches that of the noise distribution. Table 4a in the appendix demonstrates this relationship empirically.

The overall relationship between likelihood and CV with respect to noise specification in LSNMs is as follows, illustrated in Figure 2 with actual datasets:

1. (Theorem 4.2) CV and likelihood are negatively related (Figures 2a and 2b), with either correctly or incorrectly specified noise distribution form.

2. (Identifiability) When the form of the noise distribution is correctly specified, the causal model always has a higher likelihood (Figure 2a).

3. (Corollary 4.3) When the form of the noise distribution is misspecified, the anti-causal model can have a higher likelihood under misleading CVs (Figure 2b).

*ANM vs. LSNM.* The relationship (2) is in general different for ANMs, when the noise variance is misleading (see Figure 1 and Appendix P). Proposition H.1 shows that the standardization lemma Lemma 4.1 does not apply to ANMs (confirmed by empirical results in Table 4b in the appendix).

## 5 Robustness of independence testing

This section describes IT methods for cause-effect learning in LSNMs, including a new IT method based on affine flows. Theoretical results below explain why IT methods are robust to noise misspecification and misleading CVs in LSNMs.

### 5.1 The independence testing method

Inspired by the breakthrough DirectLiNGAM approach [25], independence testing has been used in existing methods for various SCMs [7, 19, 28, 9]. Like ML methods (Equation (4)), IT methods fit the model parameters in both directions, typically maximizing the data likelihood (Equation (3)). The difference is in the model selection step, where IT methods select the direction with the highest degree of independence between the fitted model residuals and the putative cause.

Algorithm 1 is the pseudo-code for our IT method **CAREFL-H**. We fit the functions $f$ and $g$ in Equation (1), with the affine flow estimator $\mathbf{T}$ from CAREFL-M [11], implemented using neural

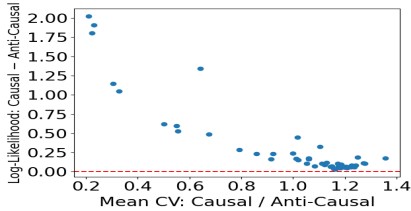

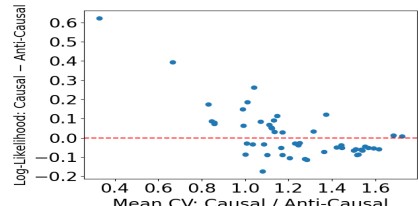

(a) Log-likelihood difference under correct noise specification: $Gaussian(0, 1)$ noise

(b) Log-likelihood difference under noise misspecification: $uniform(-1, 1)$ noise

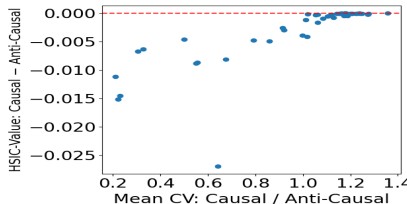

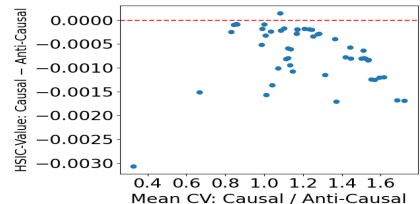

(c) HSIC difference under correct noise specification: $Gaussian(0, 1)$ noise

(d) HSIC difference under noise misspecification: $uniform(-1, 1)$ noise

Figure 2: Visualization of Table 1. First row (2a,2b): ML methods. Second row (2c,2d): IT methods. *Y-axis < 0.0*: A ML method returns the *incorrect* anti-causal direction. *Y-axis > 0.0*: An IT method returns the *incorrect* anti-causal direction. ML methods may fail under misspecification and misleading CVs (2b, when *X-axis > 1* ). IT methods are more robust (2d).

networks. For details on CAREFL-M and learning the flow transformation **T** see Appendix I. This combination of affine flow model with IT appears to be new. Another IT method is to test the independence of both residuals [6] (Appendix J). We found that this performs similarly to CAREFL-H and therefore report results only for the more common DirectLiNGAM-style method.

As in previous work [14], we use the Hilbert-Schmidt independence criterion (HSIC) [5] to measure (in)dependence throughout the paper. HSIC measures the squared distance between the joint probability of the two variables and the product of their marginals embedded in the reproducible kernel Hilbert space. We have $HSIC(U, V) = 0$ if and only if $U \perp\!\!\!\perp V$.

## 5.2 Theoretical comparison between IT and ML under noise misspecification in LSNMs

The intuition for why independence testing works is that if LSNM training is consistent, testing the independence of the putative cause and the residual will indicate the correct causal model. The next theorem provides a formal statement of this intuition.

**Theorem 5.1.** *For data pairs $(X, Y)$ generated by an LSNM model $X \to Y$, let $\widehat{f}(X)$ denote the conditional mean estimator and $\widehat{g}(X)$ denote the conditional standard deviation estimator. Under the consistency conditions that $\widehat{f}(X) = \mathbb{E}[Y|X]$ and $\widehat{g}(X) = STD[Y|X]$, the reconstructed noise is independent of the cause, i.e. $\widehat{Z}_Y \perp\!\!\!\perp X$, even under noise misspecification.*

*Proof.* According to Equation (1),

$$\widehat{f}(X) = \mathbb{E}[Y|X] = \mathbb{E}[f(X) + g(X) \cdot Z_Y|X] = f(X) + g(X) \cdot \mathbb{E}[Z_Y|X] = f(X) + g(X) \cdot \mathbb{E}[Z_Y]$$
$$\widehat{g}(X) = STD[Y|X] = STD[f(X) + g(X) \cdot Z_Y|X] = |g(X)| \cdot STD[Z_Y|X] = g(X) \cdot STD[Z_Y]$$

Thus, $\widehat{Z}_Y = \frac{Y - \widehat{f}(X)}{\widehat{g}(X)} = \frac{f(X) + g(X) \cdot Z_Y - f(X) - g(X) \cdot \mathbb{E}[Z_Y]}{g(X) \cdot STD[Z_Y]} = \frac{Z_Y - \mathbb{E}[Z_Y]}{STD[Z_Y]}$, where $\mathbb{E}[Z_Y]$ and $STD[Z_Y] > 0$ are constants. Therefore, $\widehat{Z}_Y$ and $Z_Y$ are identical up to shift and scale. Since $Z_Y \perp\!\!\!\perp X$, therefore, $\widehat{Z}_Y \perp\!\!\!\perp X$. $\qquad \square$

Note that the proof does not involve the specification of the noise prior distribution $P_{Z_Y}$, which shows that under the consistency condition, the conclusion $\widehat{Z}_Y \perp\!\!\!\perp X$ holds whether the noise prior distribution is misspecified or not.

---

**Algorithm 1** CAREFL-H

---

1: **Input:** data pairs $D := (X, Y)$, the flow estimator $\mathbf{T}$ of CAREFL-M with prior $P_Z$, and an HSIC estimator
2: **Output:** estimated causal direction $dir$
3: Split $D$ into training set $D_{train} := (X_{train}, Y_{train})$ and testing set $D_{test} := (X_{test}, Y_{test})$
4: Optimize $\mathbf{T}_{\widehat{\theta}, \widehat{\psi}, \widehat{\varsigma}}(D_{train}; P_Z)$ in $X \rightarrow Y$ direction via ML to estimate $\widehat{f}$ and $\widehat{g}$
5: Compute the residual $\widehat{Z}_Y := \frac{Y_{test} - \widehat{f}(X_{test})}{\widehat{g}(X_{test})}$
6: Optimize $\mathbf{T}_{\widehat{\theta'}, \widehat{\psi'}, \widehat{\varsigma'}}(D_{train}; P_Z)$ in $X \leftarrow Y$ direction via ML to estimate $\widehat{h}$ and $\widehat{k}$
7: Compute the residual $\widehat{Z}_X := \frac{X_{test} - \widehat{h}(Y_{test})}{\widehat{k}(Y_{test})}$
8: **if** $HSIC(X_{test}, \widehat{Z}_Y) < HSIC(Y_{test}, \widehat{Z}_X)$ **then**
9: $\quad dir := X \rightarrow Y$
10: **else if** $HSIC(X_{test}, \widehat{Z}_Y) > HSIC(Y_{test}, \widehat{Z}_X)$ **then**
11: $\quad dir := X \leftarrow Y$
12: **else**
13: $\quad dir := \textit{no conclusion}$
14: **end if**

---

In contrast, the likelihood of a causal model depends on the specification of the noise prior distribution $P_{Z_Y}$ (Equation (2)). This explains why IT methods are more robust than ML methods under noise misspecification in LSNMs.

In the context of ANMs, Mooij et al. [14] provide another sufficient condition for $\widehat{Z}_Y \perp\!\!\!\perp X$ called suitability. In Appendix K, we show how the suitability concept can be adapted for LSNMs to provide another argument for the robustness of IT methods.

## 6 Limitation of IT-based methods

Limitations of IT-based methods include the following: 1) When the noise prior distribution is correctly specified, ML-based methods can utilize this information to be more sample efficient. Conversely, since IT-based methods are less sensitive to the noise prior distribution, they require a larger sample size to infer causal direction accurately. 2) IT-based methods have higher computational cost, especially with nonlinear independence testing. 3) IT-based methods require high-capacity function estimators to model the unknown functions $f$ and $g$ for a good noise reconstruction.

Despite these complications, we advocate the use of IT-based methods over ML-based methods under noise misspecification and misleading CVs in LSNMs, because they are more reliable in such settings.

## 7 Experiments

On synthetic datasets, we find that across different hyperparameter choices the IT method (CAREFL-H) produces much higher accuracy than the ML method (CAREFL-M) in the difficult settings with noise misspecification and misleading CVs, and produces comparable accuracy in the easier settings without noise misspecification or misleading CVs. On real-world data where the ground-truth noise distribution is unknown, the IT method is also more robust across different hyperparameter choices.

For all experiments, we start with the same default hyperparameter values for both CAREFL-M and CAREFL-H and alter one value at a time. The default hyperparameter values are those specified in CAREFL-M [11] for the Tübingen Cause-Effect Pairs benchmark [14]. Please see Appendix L for more details on default and alternative hyperparameter values. Previous work [14, 9] reported that ML methods perform better with data splitting (split data into training set for model fitting and testing set for model selection) and IT methods perform better with data recycling (the same data is used for both model fitting and selection). Therefore, we use both splitting methods: (i) CAREFL(0.8): 80% as training and 20% as testing. (ii) CAREFL(1.0): training = testing = 100%. The training procedure is not supervised: no method accesses the ground-truth direction.

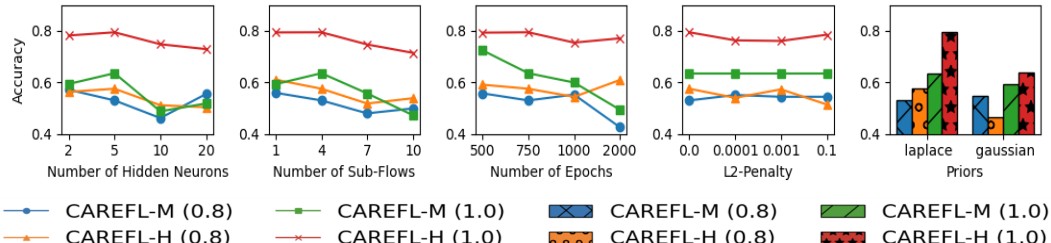

Figure 3: **Weighted accuracy** over 99 datasets from **Tübingen Cause-Effect Pairs benchmark**.

Table 2: Accuracy of methods on the SIM and Tübingen Cause-Effect Pairs benchmarks. For methods other than CAREFL-M and CAREFL-H, we use the best results reported in Immer et al. [9].

(a) Accuracy with SIM benchmarks

|  | LOCI-M | LOCI-H | GRCI | BQCD | HECI | CAM | RESIT | CAREFL-M | CAREFL-H |
|---|---|---|---|---|---|---|---|---|---|
| SIM | 0.52 | 0.79 | 0.77 | 0.62 | 0.49 | 0.57 | 0.77 | 0.55 | **0.80** |
| SIM-c | 0.50 | 0.83 | 0.77 | 0.72 | 0.55 | 0.60 | 0.82 | 0.58 | **0.85** |
| SIM-ln | 0.79 | 0.73 | 0.77 | 0.80 | 0.65 | **0.87** | **0.87** | 0.84 | 0.83 |
| SIM-G | 0.78 | 0.81 | 0.70 | 0.64 | 0.56 | 0.81 | 0.78 | **0.82** | 0.79 |

(b) Weighted accuracy with Tübingen Cause-Effect Pairs benchmark

|  | LOCI-M | LOCI-H | GRCI | BQCD | HECI | CAM | RESIT | CAREFL-M | CAREFL-H |
|---|---|---|---|---|---|---|---|---|---|
| Tübingen Cause-Effect Pairs | 0.57 | 0.60 | **0.82** | 0.77 | 0.71 | 0.58 | 0.57 | 0.73 | **0.82** |

We use a consistent HSIC estimator with Gaussian kernels [21]. A summary of experimental datasets is provided in Appendix Table 7. All the datasets are standardized to have mean 0 and variance 1. The datasets in Section 7.1 are generated by LSNMs and the datasets in Sections 7.2 and 7.3 are not. Consequently, the results demonstrate that the proposed method performs well not only when the ground-truth SCMs are strictly LSNMs, but also when the ground-truth SCMs are unknown or not strictly LSNMs.

## 7.1 Results on synthetic LSNM data

See Appendix M for the definition of the ground-truth LSNM SCMs and details on how synthetic datasets are generated from them. The sample sizes in each synthetic dataset are 500 or 5,000. As shown in Appendix Table 7, most synthetic datasets generated by such SCMs have misleading CVs. Based on the analysis of Section 4 we formulate the following hypotheses. (i) CAREFL-M should be accurate given a correct specification on the form of noise distribution, with or without misleading CVs. (ii) With noise misspecification and mild misleading CVs, the accuracy of CAREFL-M should be reduced. (iii) With noise misspecification and severe misleading CVs, the accuracy should be very low, often below 50%. Overall, the results from the experiments confirm our hypotheses.

### 7.1.1 Noise misspecification

We evaluate CAREFL-M and CAREFL-H against data generated with $uniform(-1, 1)$, $exponential(1)$, $continuousBernoulli(0.9)$ or $beta(0.5, 0.5)$ noise, covered by our identifiability Theorem C.2 (except $beta(0.5, 0.5)$). Khemakhem et al. [11] empirically find that CAREFL-M with a Laplace model prior is robust with similar ground-truth noise distributions such as Gaussian and Student's t. We show that it may fail remarkably with a dissimilar distribution.

We summarize findings from the 336 settings here; the detailed results are given in Appendix Figures 6 to 17. In 289 settings (86.01%), both CAREFL-M(0.8) and CAREFL-M(1.0) select the correct causal

direction with less than 50% random accuracy. Furthermore, in 110 settings (32.74%) both CAREFL-M(0.8) and CAREFL-M(1.0) fail catastrophically with an accuracy of 0%. These experiments also show that the accuracy of CAREFL-M often decreases as $N$ increases. In contrast, CAREFL-H(1.0) achieves better accuracy than CAREFL-M in 333 settings (99.11%). The accuracy of CAREFL-H(1.0) goes below 50% in only 6 settings (1.79%). The results demonstrate the robustness of the IT method under noise misspecification and misleading CVs, across different hyperparameter choices.

Appendix Table 7 and Appendix Figure 17 show that with severe misleading CVs, the accuracy of CAREFL-M is close to 0%. This is much lower than the corresponding cases with mild misleading CVs in Appendix Figures 9 and 13.

### 7.1.2 Correct specification on the form of the noise distribution

These experiments show that CAREFL-H is comparable with CAREFL-M under correct form of noise specification, with or without misleading CVs, especially on larger datasets, as long as the affine model capacity is sufficient. We evaluate CAREFL-M and CAREFL-H against data generated with $Gaussian(0,1)$ and $Laplace(0,1)$ noise. The detailed results are in Appendix Figures 18 to 23. We find that CAREFL-M is more sample efficient than CAREFL-H when the model prior matches the data, which is a general pattern for ML vs. IT methods [23]. Consistent with the suitability results in Appendix K.1, the accuracy of CAREFL-H improves with more data. For example, with $N = 500$, there are 49 out of 84 settings (58.33%) where CAREFL-M outperforms CAREFL-H(1.0). However, with $N = 5,000$, CAREFL-H(1.0) achieves similar accuracy as CAREFL-M on all datasets (except LSNM-sigmoid-sigmoid with $Laplace(0,1)$ noise.) In addition, CAREFL-H(1.0) may underperform CAREFL-M when the number of hidden neurons, sub-flows or training epochs is low. An IT method requires more model capacity to fit the LSNM functions and produce a good reconstruction of the noise.

## 7.2 Results on synthetic benchmarks

Similar to Tagasovska et al. [29], Immer et al. [9], we compare CAREFL-M and CAREFL-H against the SIM benchmark suite [14]. SIM comprises 4 sub-benchmarks: default (SIM), with one confounder (SIM-c), low noise levels (SIM-ln) and Gaussian noise (SIM-G). In this benchmark, most datasets do not have misleading CVs (Appendix Table 7), which favors ML methods. Each sub-benchmark contains 100 datasets and each dataset has $N = 1000$ data pairs. As shown in Appendix Figure 24, CAREFL-M and CAREFL-H(1.0) achieve similar accuracy on SIM-ln and SIM-G across different hyperparameter choices. For SIM and SIM-c, CAREFL-H, especially CAREFL-H(1.0), outperforms CAREFL-M by 20%-30% in all settings. The accuracy of CAREFL-M is only about random guess (40%-60%) on SIM and SIM-c.

Table 2a compares CAREFL-H with SOTA methods. For each CAREFL method, we report the best accuracy obtained in Appendix Figure 24 without further tuning. CAREFL-H achieves the best accuracy on SIM and SIM-c, and achieves competitive accuracy on SIM-ln and SIM-G.

## 7.3 Results on real-world benchmarks: Tübingen Cause-Effect Pairs

The Tübingen Cause-Effect Pairs benchmark [14] is commonly used to evaluate cause-effect inference algorithms [11, 30, 9]. To be consistent with previous work [29, 28, 9], we exclude 6 multivariate and 3 discrete datasets (#47, #52-#55, #70, #71, #105, #107) and utilize the remaining 99 bivariate datasets. As recommended by Mooij et al. [14], we report weighted accuracy. 40% of datasets in the benchmark feature misleading CVs. *CAREFL-H(1.0) outperforms CAREFL-M in all configurations by large margins (7%-30%)*; see Figure 3. We also compare CAREFL-H with SOTA methods (see Appendix N for hyperparameters). Table 2b shows that CAREFL-H achieves the SOTA accuracy (82%) and is 9% more accurate than CAREFL-M [11].

Furthermore, LSNM methods (i.e. LOCI, GRCI, BQCD, HECI, CAREFL) perform better than ANM methods (i.e. CAM, RESIT). The reason is that LSNM is a weaker assumption than ANM and allows for heteroscedastic noise.

## 8   Conclusion and future work

We identified a failure mode of maximum-likelihood (ML) methods for cause-effect inference in location-scale noise models (LSNMs). Our analysis shows that the failure occurs when the form of the noise distribution is misspecified and conditional variances are misleading (i.e., higher in the causal direction). Selecting causal models by independence tests (IT) is robust even in this difficult setting. Extensive empirical evaluation compared the ML method and a new IT method based on affine flows, on both synthetic and real-world datasets. The IT flow method achieves better accuracy under noise misspecification and misleading CVs, with robust performance across different hyperparameter choices. Future directions include improving the sample efficiency of IT methods, and improving the robustness of ML methods by learning the noise distribution instead of using a fixed prior.

## 9   Acknowledgments

This research was supported by a Discovery Grant from the Natural Sciences and Engineering Research Council of Canada (NSERC).

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

## A   Extension to multivariate settings

To extend a bivariate cause-effect inference method to multivariate, the most common approach is to first use the PC algorithm [27] (or other algorithms) to learn a completed partially directed acyclic graph (CPDAG) that may contain undirected edges. Then, orient the undirected edges using the cause-effect inference method. This approach can be found in Tagasovska et al. [29], Khemakhem et al. [11], Xu et al. [30], Strobl and Lasko [28].

## B   Derivation of Equation (2)

**Lemma B.1.** *For an LSNM model $X \to Y$ defined in Equation* (1)*, we have $P_{Y|X}(Y|X) = P_{Z_Y}(\frac{Y-f(X)}{g(X)}) \cdot \frac{1}{g(X)}$, where $P_{Z_Y}$ is the noise distribution.*

For the data distribution in the $X \to Y$ direction:

$$P_{\to,P_Z}(X,Y) = P_X(X) \cdot P_{Y|X}(Y|X) = P_X(X) \cdot P_{Z_Y}(\frac{Y-f(X)}{g(X)}) \cdot \frac{1}{g(X)}$$

Similarly, in the $X \leftarrow Y$ direction:

$$P_{\leftarrow,P_Z}(X,Y) = P_Y(Y) \cdot P_{X|Y}(X|Y) = P_Y(Y) \cdot P_{Z_X}(\frac{X-h(Y)}{k(Y)}) \cdot \frac{1}{k(Y)}$$

*Proof for Lemma B.1.* For an LSNM model defined in Equation (1), we have $Z_Y = \frac{Y-f(X)}{g(X)}$. Hence, $\frac{\partial Z_Y}{\partial Y} = \frac{1}{g(X)}$. Since $g(X) > 0$, so $|\frac{\partial Z_Y}{\partial Y}| = |\frac{1}{g(X)}| = \frac{1}{g(X)}$.

$$
\begin{aligned}
&P_{Y|X}(Y|X) \\
=&P_{Z_Y}(Z_Y|X) \cdot |\det \frac{\partial Z_Y}{\partial Y}| \\
&\text{(via change of variables)} \\
=&P_{Z_Y}(Z_Y) \cdot |\det \frac{\partial Z_Y}{\partial Y}| \\
=&P_{Z_Y}(\frac{Y-f(X)}{g(X)}) \cdot |\frac{\partial Z_Y}{\partial Y}| \\
=&P_{Z_Y}(\frac{Y-f(X)}{g(X)}) \cdot \frac{1}{g(X)}
\end{aligned}
$$

$\square$

## C   Identifiability of LSNMs with correct noise distribution

Strobl and Lasko [28], Immer et al. [9] prove the identifiability of LSNMs, assuming a correctly specified noise distribution. That is, given that the data generating distribution $(X,Y)$ follows an LSNM in the direction $X \to Y$, the same distribution with equal likelihood cannot be induced by an LSNM in the backward direction $X \leftarrow Y$, except in some pathological cases. In terms of our notation, direction identifiability means that if $(\to, P_Z)$ is the data generating model, then

$$P([L_{\to,P_Z}(D) - L_{\leftarrow,P_Z}(D)] > 0) \to 1 \text{ as } N \to \infty$$

Immer et al. [9] prove the following identifiability result:

**Theorem C.1** (Theorem 1 from [9]). *For data $(X,Y)$ that follows an LSNM in both direction $X \to Y$ and $X \leftarrow Y$, i.e.,*

$$
\begin{aligned}
Y &= f(X) + g(X) \cdot Z_Y, \text{ where } X \perp\!\!\!\perp Z_Y \\
X &= h(Y) + k(Y) \cdot Z_X, \text{ where } Y \perp\!\!\!\perp Z_X
\end{aligned}
$$

*The following condition must be true:*

$$(\log p(y))'' + \frac{g'(x)}{G(x,y)} \cdot (\log p(y))' + \frac{\partial^2}{\partial y^2} \cdot \nu_{X|Y}(x|y) + \frac{g(x)}{G(x,y)} \cdot \frac{\partial^2}{\partial y \partial x} \cdot \nu_{X|Y}(x|y)$$
$$+ \frac{g'(x)}{G(x,y)} \cdot \frac{\partial}{\partial y} \cdot \nu_{X|Y}(x|y) = 0 \tag{5}$$

*where $G(x,y) = g(x) \cdot f'(x) + g'(x) \cdot [y - f(x)] \neq 0$ and $\nu_{X|Y}(x|y) = \log p_{Z_X}(\frac{x-h(y)}{k(y)}) - \log k(y)$.*

They state that Equation (5) will be false except for "pathological cases". In addition, Khemakhem et al. [11] provide sufficient conditions for LSNMs with Gaussian noise to be identifiable. Our next theorem provides identifiability results for some non-Gaussian noise distributions:

**Theorem C.2.** *Suppose that the true data-generating distribution follows an LSNM model in both $X \to Y$ and $X \leftarrow Y$ directions:*

1. *If the noise distribution is $uniform(a, b)$, then both $g(X)$ and $k(Y)$ are constant functions.*

2. *If the noise distribution is $continuousBernoulli(\lambda \neq 0.5)^2$ or $exponential(\lambda)$, then one of the following conditions holds:*

   - $\frac{1}{g(X)}$ *and* $\frac{1}{k(Y)}$ *are constant functions.*
   - $\frac{1}{g(X)}$ *and* $\frac{1}{k(Y)}$ *are linear functions with the same coefficients on $X$ and $Y$, respectively.*

The proof is in Appendix D. Essentially, the theorem shows that for uniform, exponential, and continuousBernoulli noise distributions, the true LSNM model can be identified unless it degenerates to (i) a homoscedastic additive noise model or (ii) a heteroscedastic model with the same linear scale in both directions.

## D Identifiability proofs

In this section, we prove Theorem C.2.

If the data $(x, y)$ follows an LSNM in the forward (i.e. causal) model:

$$y := f(x) + g(x) \cdot z_Y$$

where $z_Y$ is the noise term, $x \perp\!\!\!\perp z_Y$, $g(x) > 0$ for all $x$ on its domain. We assume $f(\cdot)$ and $g(\cdot)$ are twice-differentiable on the domain of $x$.

If the data $(x, y)$ follows an LSNM in the backward (i.e. anti-causal) model:

$$x := h(y) + k(y) \cdot m_X$$

where $m_X$ is the noise term, $y \perp\!\!\!\perp m_X$, $k(y) > 0$ for all $y$ on its domain. We assume $h(\cdot)$ and $k(\cdot)$ are twice-differentiable on the domain of $y$.

$z_Y$ and $m_X$ follow one of $uniform(a, b)$, $exponential(\lambda)$ or $continuousBernoulli(\lambda)$ distribution accordingly.

*Proof for $uniform(a, b)$ noise.* For the causal model, according to Lemma B.1, we have $P_{Y|X}(y|x) = P_{Z_Y}(\frac{y-f(x)}{g(x)}) \cdot \frac{1}{g(x)} = P_{Z_Y}(z_Y) \cdot \frac{1}{g(x)} = \frac{1}{b-a} \cdot \frac{1}{g(x)} = \frac{1}{(b-a) \cdot g(x)}$. Similarly, for the backward model, we have $P_{X|Y}(x|y) = \frac{1}{(b-a) \cdot k(y)}$.

The joint likelihood of the observation $(x, y)$ in the causal model is:

$$P_{\to, P_{Z_Y}}(x, y) = P_X(x) \cdot P_{Y|X}(y|x) = P_X(x) \cdot \frac{1}{(b-a) \cdot g(x)}$$

---

[2]The case of $continuousBernoulli(\lambda = 0.5)$ is equivalent to $uniform(0, 1)$.

The joint likelihood of the observation $(x, y)$ in the backward model is:

$$P_{\leftarrow, P_{M_X}}(x, y) = P_Y(y) \cdot P_{X|Y}(x|y) = P_Y(y) \cdot \frac{1}{(b-a) \cdot k(y)}$$

If the data follows both models:

$$P_{\rightarrow, P_{Z_Y}}(x, y) = P_{\leftarrow, P_{M_X}}(x, y)$$

$$P_X(x) \cdot \frac{1}{(b-a) \cdot g(x)} = P_Y(y) \cdot \frac{1}{(b-a) \cdot k(y)}$$

$$P_X(x) \cdot \frac{1}{g(x)} = P_Y(y) \cdot \frac{1}{k(y)}$$

Take the derivative of both sides with respect to $x$:

$$P_X(x) \cdot \left(\frac{1}{g(x)}\right)' + (P_X(x))' \cdot \frac{1}{g(x)} = 0$$

$$P_X(x) \cdot -1 \cdot \frac{1}{g(x)^2} \cdot g(x)' + 0 \cdot \frac{1}{g(x)} = 0$$

$$P_X(x) \cdot -1 \cdot \frac{1}{g(x)^2} \cdot g(x)' = 0$$

$$P_X(x) > 0$$

$$\frac{1}{g(x)^2} > 0$$

*Therefore, $g(x)' = 0$*

Similarly, if we take the derivative of both sides with respect to $y$ instead, we have:

$$k(y)' = 0$$

These imply that both $g(x)$ and $k(y)$ are constant functions. $\qquad\square$

*Proof for exponential$(\lambda)$ noise.* For the causal model, according to Lemma B.1, we have $P_{Y|X}(y|x) = P_{Z_Y}\left(\frac{y-f(x)}{g(x)}\right) \cdot \frac{1}{g(x)} = \lambda \cdot e^{-\lambda \cdot \frac{y-f(x)}{g(x)}} \cdot \frac{1}{g(x)} = \frac{\lambda}{g(x)} \cdot e^{-\frac{\lambda}{g(x)} \cdot (y-f(x))}$. Similarly, for the backward model, we have $P_{X|Y}(x|y) = \frac{\lambda}{k(y)} \cdot e^{-\frac{\lambda}{k(y)} \cdot (x-h(y))}$.

The joint likelihood of the observation $(x, y)$ in the causal model is:

$$P_{\rightarrow, P_{Z_Y}}(x, y) = P_X(x) \cdot P_{Y|X}(y|x) = P_X(x) \cdot \frac{\lambda}{g(x)} \cdot e^{-\frac{\lambda}{g(x)} \cdot (y-f(x))}$$

$$\log P_{\rightarrow, P_{Z_Y}}(x, y) = \log P_X(x) + \log \lambda - \log g(x) - \frac{\lambda}{g(x)} \cdot (y - f(x))$$

The joint likelihood of the observation $(x, y)$ in the backward model is:

$$P_{\leftarrow, P_{M_X}}(x, y) = P_Y(y) \cdot P_{X|Y}(x|y) = P_Y(y) \cdot \frac{\lambda}{k(y)} \cdot e^{-\frac{\lambda}{k(y)} \cdot (x-h(y))}$$

$$\log P_{\leftarrow, P_{M_X}}(x, y) = \log P_Y(y) + \log \lambda - \log k(y) - \frac{\lambda}{k(y)} \cdot (x - h(y))$$

If the data follows both models:

$$\log P_{\to, P_{Z_Y}}(x, y) = \log P_{\leftarrow, P_{M_X}}(x, y)$$

$$\log P_X(x) + \log \lambda - \log g(x) - \frac{\lambda}{g(x)} \cdot (y - f(x))$$

$$= \log P_Y(y) + \log \lambda - \log k(y) - \frac{\lambda}{k(y)} \cdot (x - h(y))$$

$$\log P_X(x) - \log g(x) - \lambda \cdot \frac{1}{g(x)} \cdot y + \lambda \cdot \frac{1}{g(x)} \cdot f(x)$$

$$= \log P_Y(y) - \log k(y) - \lambda \cdot \frac{1}{k(y)} \cdot x + \lambda \cdot \frac{1}{k(y)} \cdot h(y)$$

Take the derivative of both sides with respect to $x$:

$$(\log P_X(x))' - (\log g(x))' - \lambda \cdot y \cdot (\frac{1}{g(x)})' + \lambda \cdot (\frac{1}{g(x)} \cdot f(x))' = -\lambda \cdot \frac{1}{k(y)}$$

Take the derivative of both sides with respect to $y$:

$$-\lambda \cdot (\frac{1}{g(x)})' = -\lambda \cdot (\frac{1}{k(y)})'$$

$$\frac{\partial \frac{1}{g(x)}}{\partial x} = \frac{\partial \frac{1}{k(y)}}{\partial y}$$

They can be equal only if both sides are constants. Therefore, $\frac{1}{g(x)}$ and $\frac{1}{k(y)}$ are both constants or both linear functions with the same coefficient on $x$ and $y$, respectively. $\qquad\square$

*Proof for continuousBernoulli($\lambda \neq 0.5$) noise.* Please refer to *uniform* for *continuousBernoulli($\lambda = 0.5$)*, which equals to $uniform(0, 1)$. For the causal model, according to Lemma B.1, we have $P_{Y|X}(y|x) = P_{Z_Y}(\frac{y - f(x)}{g(x)}) \cdot \frac{1}{g(x)} = C_\lambda \cdot \lambda^{\frac{y - f(x)}{g(x)}} \cdot (1 - \lambda)^{1 - \frac{y - f(x)}{g(x)}} \cdot \frac{1}{g(x)}$, where $C_\lambda$ is the normalizing constant of the continuous Bernoulli distribution. Similarly, for the backward model, we have $P_{X|Y}(x|y) = C_\lambda \cdot \lambda^{\frac{x - h(y)}{k(y)}} \cdot (1 - \lambda)^{1 - \frac{x - h(y)}{k(y)}} \cdot \frac{1}{k(y)}$.

The joint likelihood of the observation $(x, y)$ in the causal model is:

$$P_{\to, P_{Z_Y}}(x, y) = P_X(x) \cdot P_{Y|X}(y|x) = P_X(x) \cdot \frac{1}{g(x)} \cdot C_\lambda \cdot \lambda^{\frac{y - f(x)}{g(x)}} \cdot (1 - \lambda)^{1 - \frac{y - f(x)}{g(x)}}$$

$$\log P_{\to, P_{Z_Y}}(x, y) = \log P_X(x) - \log g(x) + \log C_\lambda + \log \lambda^{\frac{y - f(x)}{g(x)}} + \log(1 - \lambda)^{1 - \frac{y - f(x)}{g(x)}}$$

The joint likelihood of the observation $(x, y)$ in the backward model is:

$$P_{\leftarrow, P_{M_X}}(x, y) = P_Y(y) \cdot P_{X|Y}(x|y) = P_Y(y) \cdot \frac{1}{k(y)} \cdot C_\lambda \cdot \lambda^{\frac{x - h(y)}{k(y)}} \cdot (1 - \lambda)^{1 - \frac{x - h(y)}{k(y)}}$$

$$\log P_{\leftarrow, P_{M_X}}(x, y) = \log P_Y(y) - \log k(y) + \log C_\lambda + \log \lambda^{\frac{x - h(y)}{k(y)}} + \log(1 - \lambda)^{1 - \frac{x - h(y)}{k(y)}}$$

If the data follows both models:

$$\log P_{\rightarrow, P_{Z_Y}}(x, y) = \log P_{\leftarrow, P_{M_X}}(x, y)$$

$$\log P_X(x) - \log g(x) + \log C_\lambda + \log \lambda^{\frac{y - f(x)}{g(x)}} + \log(1 - \lambda)^{1 - \frac{y - f(x)}{g(x)}}$$

$$= \log P_Y(y) - \log k(y) + \log C_\lambda + \log \lambda^{\frac{x - h(y)}{k(y)}} + \log(1 - \lambda)^{1 - \frac{x - h(y)}{k(y)}}$$

$$\log P_X(x) - \log g(x) + \log C_\lambda + \frac{y - f(x)}{g(x)} \cdot \log \lambda + (1 - \frac{y - f(x)}{g(x)}) \cdot \log(1 - \lambda)$$

$$= \log P_Y(y) - \log k(y) + \log C_\lambda + \frac{x - h(y)}{k(y)} \cdot \log \lambda + (1 - \frac{x - h(y)}{k(y)}) \cdot \log(1 - \lambda)$$

$$\log P_X(x) - \log g(x) + \log C_\lambda + \frac{1}{g(x)} \cdot \log \lambda \cdot y - \frac{1}{g(x)} \cdot$$

$$\log \lambda \cdot f(x) + \log(1 - \lambda) - \log(1 - \lambda) \cdot \frac{1}{g(x)} \cdot y + \log(1 - \lambda) \cdot \frac{1}{g(x)} \cdot f(x)$$

$$= \log P_Y(y) - \log k(y) + \log C_\lambda + \frac{1}{k(y)} \cdot \log \lambda \cdot x - \frac{1}{k(y)} \cdot$$

$$\log \lambda \cdot h(y) + \log(1 - \lambda) - \log(1 - \lambda) \cdot \frac{1}{k(y)} \cdot x + \log(1 - \lambda) \cdot \frac{1}{k(y)} \cdot h(y)$$

Take the derivative of both sides with respect to $x$:

$$(\log P_X(x))' - \frac{1}{g(x)} \cdot g(x)' - 1 \cdot \frac{1}{g(x)^2} \cdot g(x)' \cdot \log \lambda \cdot y$$

$$- (\frac{1}{g(x)} \cdot \log \lambda \cdot f(x))' - \log(1 - \lambda) \cdot -1 \cdot \frac{1}{g(x)^2} \cdot g(x)' \cdot y$$

$$+ (\log(1 - \lambda) \cdot \frac{1}{g(x)} \cdot f(x))' = \frac{1}{k(y)} \cdot \log \lambda - \log(1 - \lambda) \cdot \frac{1}{k(y)}$$

Take the derivative of both sides with respect to $y$:

$$- \frac{1}{g(x)^2} \cdot g(x)' \cdot \log \lambda - \log(1 - \lambda) \cdot -1 \cdot \frac{1}{g(x)^2} \cdot g(x)'$$

$$= -1 \cdot \frac{1}{k(y)^2} \cdot k(y)' \cdot \log \lambda - \log(1 - \lambda) \cdot -1 \cdot \frac{1}{k(y)^2} \cdot k(y)'$$

$$\frac{1}{g(x)^2} \cdot g(x)' \cdot \log \lambda - \log(1 - \lambda) \cdot \frac{1}{g(x)^2} \cdot g(x)'$$

$$= \frac{1}{k(y)^2} \cdot k(y)' \cdot \log \lambda - \log(1 - \lambda) \cdot \frac{1}{k(y)^2} \cdot k(y)'$$

$$\frac{1}{g(x)^2} \cdot g(x)' \cdot (\log \lambda - \log(1 - \lambda)) = \frac{1}{k(y)^2} \cdot k(y)' \cdot (\log \lambda - \log(1 - \lambda))$$

Since $\lambda \neq 0.5$, therefore $\log \lambda - \log(1 - \lambda) \neq 0$

$$\frac{1}{g(x)^2} \cdot g(x)' = \frac{1}{k(y)^2} \cdot k(y)'$$

$$\frac{\partial \frac{1}{g(x)}}{\partial x} = \frac{\partial \frac{1}{k(y)}}{\partial y}$$

They can be equal only if both sides are constants. Therefore, $\frac{1}{g(x)}$ and $\frac{1}{k(y)}$ are both constants or both linear functions with the same coefficient on $x$ and $y$, respectively. $\qquad\square$

Table 3: Table 1 without dataset standardization.

| True Noise | $Gaussian(0, 1)$ | | | | | $Uniform(-1, 1)$ | | | | |
|---|---|---|---|---|---|---|---|---|---|---|
| $\alpha$ | 0.1 | 0.5 | 1 | 5 | 10 | 0.1 | 0.5 | 1 | 5 | 10 |
| $\overline{\mathbb{V}}[Y\|X]$ | 0.106 | 0.846 | 3.162 | 77.309 | 309.035 | 0.013 | 0.199 | 0.780 | 19.386 | 77.529 |
| $\overline{\mathbb{V}}[X\|Y]$ | 0.446 | 0.707 | 0.790 | 0.818 | 0.814 | 0.016 | 0.126 | 0.189 | 0.228 | 0.226 |
| Percentage of Datasets With Misleading CVs | 0% | 90% | 100% | 100% | 100% | 40% | 80% | 100% | 100% | 100% |
| CAREFL-M | 1.0 | 1.0 | 1.0 | 1.0 | 1.0 | 0.6 | 0.5 | 0.5 | 0.0 | 0.0 |
| CAREFL-H | 1.0 | 1.0 | 1.0 | 1.0 | 1.0 | 1.0 | 1.0 | 1.0 | 1.0 | 0.9 |

# E   Proof for Lemma 4.1

*Proof.* According to Equation (2), we have $P_{\rightarrow, P_Z}(Y|X; f, g) = P_{Z_Y}(\frac{Y - f(X)}{g(X)}) \cdot \frac{1}{g(X)}$ and $P_{\rightarrow, P_{Z'}}(Y|X; f', g') = P_{Z'_Y}(\frac{Y - f'(X)}{g'(X)}) \cdot \frac{1}{g'(X)}$. Let $Z'_Y$ be the standardized version of $Z_Y$, i.e. $Z'_Y = \frac{Z_Y - \mu}{\sigma}$.

$$Z_Y = Z'_Y \cdot \sigma + \mu$$
$$\frac{\partial Z_Y}{\partial Z'_Y} = \sigma$$
$$P_{Z'_Y}(\frac{Y - f'(X)}{g'(X)}) \cdot \frac{1}{g'(X)} = P_{Z_Y}(\frac{Y - f'(X)}{g'(X)} \cdot \sigma + \mu) \cdot |\det \frac{\partial Z_Y}{\partial Z'_Y}| \cdot \frac{1}{g'(X)}$$
$$= P_{Z_Y}(\frac{Y - f'(X)}{g'(X)} \cdot \sigma + \mu) \cdot \sigma \cdot \frac{1}{g'(X)}$$

Let $\frac{1}{g(X)} = \sigma \cdot \frac{1}{g'(X)}$ and $f'(X) = f(X) + \mu \cdot g(X)$, then

$$g'(X) = \sigma \cdot g(X)$$
$$P_{Z_Y}(\frac{Y - f'(X)}{g'(X)} \cdot \sigma + \mu) \cdot \sigma \cdot \frac{1}{g'(X)} = P_{Z_Y}(\frac{Y - (f(X) + \mu \cdot g(X))}{\sigma \cdot g(X)} \cdot \sigma + \mu) \cdot \sigma \cdot \frac{1}{\sigma \cdot g(X)}$$
$$= P_{Z_Y}(\frac{Y - f(X)}{g(X)}) \cdot \frac{1}{g(X)}$$

Therefore, $P_{\rightarrow, P_Z}(Y|X; f, g) = P_{\rightarrow, P_{Z'}}(Y|X; f', g')$.

For conditional variances, we have:

$$\mathbb{V}_{Z_Y}[Y|X] = g^2(X) \cdot \mathbb{V}[Z_Y] = g^2(X) \cdot \sigma^2$$
$$\mathbb{V}_{Z'_Y}[Y|X] = g'^2(X) \cdot \mathbb{V}[Z'_Y] = g'^2(X) \cdot 1 = \sigma^2 \cdot g^2(X)$$

Therefore, $\mathbb{V}_{Z_Y}[Y|X] = \mathbb{V}_{Z'_Y}[Y|X]$. $\qquad\square$

# F   Proof for Theorem 4.2

*Proof.* An LSNM model defined in Equation (1) entails the following relationships:

$$\mathbb{V}[Y|X] = g^2(X) \cdot \mathbb{V}[Z_Y] = g^2(X)$$
$$\log P_{\rightarrow, P_Z}(Y|X) = \log P_{Z_Y}(Z_Y) + \log \frac{1}{g(X)}$$

where Lemma 4.1 shows that we can always assume $\mathbb{V}[Z_Y] = 1$. Also, since there is no causal relationship between $X$ and $Z_Y$ in the ground-truth data, i.e., changing $X$ does not change $Z_Y$, we have $\frac{\partial Z_Y}{\partial X} = 0$, $\frac{\partial \log P_{Z_Y}(Z_Y)}{\partial X} = 0$ and $\frac{\partial \log P_{Z_Y}(Z_Y)}{\partial g^2(X)} = 0$.

$$\frac{\partial \log P_{\to, P_Z}(Y|X)}{\partial \mathbb{V}[Y|X]} = \frac{\partial \log P_{Z_Y}(Z_Y)}{\partial g^2(X)} + \frac{\partial \log \frac{1}{g(X)}}{\partial g^2(X)}$$

$$= 0 + \frac{\partial \log \frac{1}{g(X)}}{\partial g^2(X)}$$

$$= -\frac{1}{2 \cdot g^2(X)}$$

Since $g(X) > 0$, i.e., strictly positive on the domain of $X$, therefore, $\frac{\partial \log P_{\to, P_Z}(Y|X)}{\partial \mathbb{V}[Y|X]} < 0$.

$\square$

## G  Proof for Corollary 4.3

*Proof.* To compare the data log-likelihood under the causal model $\to$ and under the anti-causal model $\leftarrow$, we have:

$$\log P_{\to, P_Z}(X, Y) - \log P_{\leftarrow, P_Z}(X, Y)$$
$$= [\log P_{\to, P_Z}(X) + \log P_{\to, P_Z}(Y|X)] - [\log P_{\leftarrow, P_Z}(Y) + \log P_{\leftarrow, P_Z}(X|Y)]$$

Theorem 4.2 shows that conditional likelihood and conditional variance are negatively related. Therefore, by increasing $\mathbb{V}[Y|X]$ and decreasing $\mathbb{V}[X|Y]$ in the data, $\log P_{\to, P_Z}(Y|X)$ decreases and $\log P_{\leftarrow, P_Z}(X|Y)$ increases. Therefore, $\log P_{\to, P_Z}(X, Y)$ decreases, $\log P_{\leftarrow, P_Z}(X, Y)$ increases, and their difference decreases. $\square$

## H  The ANM version of Lemma 4.1

**Proposition H.1.** *Let $M_1$ be an ANM model $Y := f(X) + Z_Y$, where $Z_Y$ is the noise with mean $\mu$ and variance $\sigma^2 \neq 0$. Let $M_2$ be another ANM model $Y := f'(X) + Z'_Y$, where $Z'_Y$ is the standardized version of $Z_Y$, i.e. $Z'_Y = \frac{Z_Y - \mu}{\sigma}$. When both $Z_Y$ and $Z'_Y$ are Gaussian, $P_{\to, P_Z}(Y|X; f) \neq P_{\to, P_{Z'}}(Y|X; f')$.*

*Proof.* ANM is a restricted case of LSNM with $g(X) = 1$. Similar to Equation (2), we have $P_{\to, P_Z}(Y|X; f) = P_{Z_Y}(Y - f(X))$ for $M_1$ and $P_{\to, P_{Z'}}(Y|X; f') = P_{Z'_Y}(Y - f'(X))$ for $M_2$.

We show that $P_{Z_Y}(Y - f(X)) \neq P_{Z'_Y}(Y - f'(X))$ when $Z_Y$ and $Z'_Y$ are Gaussian.

$$P_{Z_Y}(Y - f(X)) = \frac{1}{\sigma \cdot \sqrt{2\pi}} \cdot e^{-\frac{1}{2} \cdot (\frac{Y - f(X) - \mu}{\sigma})^2}$$

$$= \frac{1}{\sigma \cdot \sqrt{2\pi}} \cdot e^{-\frac{1}{2} \cdot (\frac{Z_Y - \mu}{\sigma})^2}$$

$$= \frac{1}{\sigma \cdot \sqrt{2\pi}} \cdot e^{-\frac{1}{2} \cdot (Z'_Y)^2}$$

$$P_{Z'_Y}(Y - f'(X)) = \frac{1}{1 \cdot \sqrt{2\pi}} \cdot e^{-\frac{1}{2} \cdot (\frac{Y - f'(X) - 0}{1})^2}$$

$$= \frac{1}{\sqrt{2\pi}} \cdot e^{-\frac{1}{2} \cdot (Z'_Y)^2}$$

Therefore, $P_{\to, P_Z}(Y|X; f) \neq P_{\to, P_{Z'}}(Y|X; f')$ in the case of Gaussian unless $\sigma = 1$. Empirical results are provided in Table 4b and Appendix P. $\square$

Table 4: Comparison between the accuracy and ground-truth noise variance in LSNMs and ANMs. $N = 10,000$. We used $N(0, 1)$ as model prior. All the datasets are standardized to have mean 0 and variance 1. As implied by Lemma 4.1, ML model selection in LSNMs is insensitive to noise variance, when the form of the noise distribution is correctly specified. Proposition H.1 shows that ML model selection in ANMs is sensitive to noise variance, even when the form of the noise distribution is correctly specified.

(a) **Accuracy** over 10 datasets generated by SCM **LSNM-sine-tanh** (definition in Appendix M).

| True Noise | $N(0,1)$ | $N(0,4)$ | $N(0,25)$ | $N(0,100)$ | $N(0,400)$ |
|---|---|---|---|---|---|
| $\overline{\mathbb{V}}[Y|X]$ | 0.834 | 0.950 | 0.997 | 0.999 | 0.999 |
| $\overline{\mathbb{V}}[X|Y]$ | 0.793 | 0.810 | 0.784 | 0.781 | 0.780 |
| Percentage of Datasets With Misleading CVs | 70% | 100% | 100% | 100% | 100% |
| CAREFL-M | 1.0 | 1.0 | 1.0 | 1.0 | 1.0 |
| LOCI-M | 1.0 | 1.0 | 1.0 | 1.0 | 1.0 |

(b) **Accuracy** over 10 datasets generated by SCM **ANM-sine** (definition in Appendix P).

| True Noise | $N(0,1)$ | $N(0,4)$ | $N(0,25)$ | $N(0,100)$ | $N(0,400)$ |
|---|---|---|---|---|---|
| $\overline{\mathbb{V}}[Y|X]$ | 0.632 | 0.865 | 0.991 | 0.999 | 0.999 |
| $\overline{\mathbb{V}}[X|Y]$ | 0.756 | 0.979 | 0.999 | 0.999 | 0.999 |
| Percentage of Datasets With Misleading CVs | 0% | 0% | 0% | 50% | 40% |
| CAREFL-ANM-M | 1.0 | 1.0 | 1.0 | 0.6 | 0.5 |
| CAM | 1.0 | 1.0 | 0.7 | 0.5 | 0.6 |

# I CAREFL-M

CAREFL-M [11] models an LSNM in Equation (1) via affine flows $\mathbf{T}$. Each sub-flow $T_k \in \mathbf{T}$ is defined as the following:

$$X = t_1 + e^{s_1} \cdot Z_X$$
$$Y = t_2(X) + e^{s_2(X)} \cdot Z_Y \tag{6}$$

where $X$ is the putative cause and $Y$ is the putative effect in $X \to Y$ direction. $t_1$ and $s_1$ are constants. $t_2$ and $s_2$ are functions parameterized using neural networks. Without loss of generality, $X$ is assumed to be a function of latent noise variable $Z_X$. If $t_1 = 0$ and $s_1 = 0$, then $X = Z_X$. The exponential function $e$ ensures the multipliers to $Z$ are positive without expression loss. Similarly, for the backward direction $X \leftarrow Y$:

$$Y = t'_1 + e^{s'_1} \cdot Z_Y$$
$$X = t'_2(Y) + e^{s'_2(Y)} \cdot Z_X \tag{7}$$

where $Y$ is the putative cause and $X$ is the putative effect in $X \leftarrow Y$ direction. $t'_1$ and $s'_1$ are constants. $t'_2$ and $s'_2$ are functions parameterized using neural networks.

Given Equation (6), the joint log-likelihood of $(x, y)$ in $X \to Y$ direction is:

$$\log P_{\to, P_Z}(x, y)$$
$$= \log P_{Z_X}\left(e^{-s_1} \cdot (x - t_1)\right) + \log P_{Z_Y}\left(e^{-s_2(x)} \cdot (y - t_2(x))\right) - s_1 - s_2(x) \tag{8}$$

Similarly for the $X \leftarrow Y$ direction. Note that the priors $P_Z = \{P_{Z_X}, P_{Z_Y}\}$ in Equation (8) may mismatch the unknown ground-truth noise distribution $P_Z^* = \{P_{Z_X}^*, P_{Z_Y}^*\}$. Both CAREFL-M and

---
**Algorithm 2** CAREFL-M
---
1: **Input:** data pairs $D := (X, Y)$, and the flow estimator $\mathbf{T}$ with prior $P_Z$
2: **Output:** estimated causal direction $dir$
3: Split $D$ into training set $D_{train} := (X_{train}, Y_{train})$ and testing set $D_{test} := (X_{test}, Y_{test})$
4: Optimize $\mathbf{T}_{\hat{t}_1, \hat{s}_1, \hat{t}_2, \hat{s}_2}(D_{train}; P_Z)$ in $X \to Y$ direction via ML
5: Compute the likelihood $\widehat{L}_{\to, P_Z}(D_{test}; \mathbf{T}_{\hat{t}_1, \hat{s}_1, \hat{t}_2, \hat{s}_2})$ in $X \to Y$ direction
6: Optimize $\mathbf{T}_{\hat{t'}_1, \hat{s'}_1, \hat{t'}_2, \hat{s'}_2}(D_{train}; P_Z)$ in $X \leftarrow Y$ direction via ML
7: Compute the likelihood $\widehat{L}_{\leftarrow, P_Z}(D_{test}; \mathbf{T}_{\hat{t'}_1, \hat{s'}_1, \hat{t'}_2, \hat{s'}_2})$ in $X \leftarrow Y$ direction
8: **if** $\widehat{L}_{\to, P_Z}(D_{test}; \mathbf{T}_{\hat{t}_1, \hat{s}_1, \hat{t}_2, \hat{s}_2}) > \widehat{L}_{\leftarrow, P_Z}(D_{test}; \mathbf{T}_{\hat{t'}_1, \hat{s'}_1, \hat{t'}_2, \hat{s'}_2})$ **then**
9:      $dir := X \to Y$
10: **else if** $\widehat{L}_{\to, P_Z}(D_{test}; \mathbf{T}_{\hat{t}_1, \hat{s}_1, \hat{t}_2, \hat{s}_2}) < \widehat{L}_{\leftarrow, P_Z}(D_{test}; \mathbf{T}_{\hat{t'}_1, \hat{s'}_1, \hat{t'}_2, \hat{s'}_2})$ **then**
11:      $dir := X \leftarrow Y$
12: **else**
13:      $dir := \textit{no conclusion}$
14: **end if**
---

---
**Algorithm 3** CAREFL-H (Between Residuals)
---
1: **Input:** data pairs $D := (X, Y)$, the flow estimator $\mathbf{T}$ of CAREFL-M with prior $P_Z$, and an HSIC estimator
2: **Output:** estimated causal direction $dir$
3: Split $D$ into training set $D_{train} := (X_{train}, Y_{train})$ and testing set $D_{test} := (X_{test}, Y_{test})$
4: Optimize $\mathbf{T}_{\hat{t}_1, \hat{s}_1, \hat{t}_2, \hat{s}_2}(D_{train}; P_Z)$ in $X \to Y$ direction via ML to estimate $\hat{t}_1, \hat{s}_1, \hat{t}_2$ and $\hat{s}_2$
5: Compute the residuals $\widehat{Z}_{X, \to} := \frac{X_{test} - \hat{t}_1}{e^{\hat{s}_1}}$ and $\widehat{Z}_{Y, \to} := \frac{Y_{test} - \hat{t}_2(X)}{e^{\hat{s}_2(X)}}$
6: Optimize $\mathbf{T}_{\hat{t'}_1, \hat{s'}_1, \hat{t'}_2, \hat{s'}_2}(D_{train}; P_Z)$ in $X \leftarrow Y$ direction via ML to estimate $\hat{t'}_1, \hat{s'}_1, \hat{t'}_2$ and $\hat{s'}_2$
7: Compute the residuals $\widehat{Z}_{Y, \leftarrow} := \frac{Y_{test} - \hat{t'}_1}{e^{\hat{s'}_1}}$ and $\widehat{Z}_{X, \leftarrow} := \frac{X_{test} - \hat{t'}_2(Y)}{e^{\hat{s'}_2(Y)}}$
8: **if** $HSIC(\widehat{Z}_{X, \to}, \widehat{Z}_{Y, \to}) < HSIC(\widehat{Z}_{X, \leftarrow}, \widehat{Z}_{Y, \leftarrow})$ **then**
9:      $dir := X \to Y$
10: **else if** $HSIC(\widehat{Z}_{X, \to}, \widehat{Z}_{Y, \to}) > HSIC(\widehat{Z}_{X, \leftarrow}, \widehat{Z}_{Y, \leftarrow})$ **then**
11:      $dir := X \leftarrow Y$
12: **else**
13:      $dir := \textit{no conclusion}$
14: **end if**
---

CAREFL-H optimize Equation (8) for each direction over the training set. For CAREFL-M, it chooses the direction with ML Score $L$ over the testing set as the estimated causal direction. Detailed procedure for CAREFL-M is given in Algorithm 2. To map the parameters $\theta$, $\psi$, $\zeta$, $\theta'$, $\psi'$ and $\zeta'$ in LSNM (Equation (2)) to the flow estimator $\mathbf{T}$ of CAREFL (Equations (6) and (7)), we have $f_\theta \equiv t_2$, $g_\psi \equiv e^{s_2}$, $P_{X, \zeta} \equiv \{t_1, e^{s_1}\}$, $f_{\theta'} \equiv t'_2$, $g_{\psi'} \equiv e^{s'_2}$ and $P_{Y, \zeta'} \equiv \{t'_1, e^{s'_1}\}$.

## J   CAREFL-H alternative independence testing

In Algorithm 1, CAREFL-H tests independence between the putative cause and the residual of the putative effect in each direction, i.e., between $X$ and $\widehat{Z}_Y$ in $X \to Y$ direction, and between $Y$ and $\widehat{Z}_X$ in $X \leftarrow Y$ direction. An alternative way of testing independence is to test between the residual of the putative cause and the residual of the putative effect [6], i.e., between $\widehat{Z}_X$ and $\widehat{Z}_Y$ in both directions. Please see Algorithm 3 for complete steps.

Although in our experiments the two algorithms often produce the same estimation of causal direction, we prefer Algorithm 1, since it relies on few estimations of the residuals.

# K    Suitability theory

With *a consistent HSIC estimator*, Mooij et al. [14] show that an IT method consistently selects the causal direction for ANMs if *the regression method is suitable*. A regression method is suitable if the expected mean squared error between the predicted residuals $\widehat{E}$ and the true residuals $E$ approaches 0 in the limit of $N \to \infty$:

$$\lim_{N \to \infty} \mathbb{E}_{D,D'} \left[ \frac{1}{N} ||\widehat{E}_{1...N} - E_{1...N}||^2 \right] = 0 \tag{9}$$

where $D$ and $D'$ denote training set and testing set, respectively. Hence, with enough data a suitable regression method reconstructs the ground-truth noise.

If an HSIC estimator is consistent, the estimated HSIC value converges in probability to the population HSIC value.

$$\widehat{HSIC}(X,Y) \xrightarrow{P} HSIC(X,Y).$$

With a consistent HSIC estimator and a suitable regression method, the consistency result for ANMs in Mooij et al. [14] extends naturally to LSNMs (see Appendix K.2 for a proof outline).

**Proposition K.1.** *For identifiable LSNMs with an independent noise term in one causal direction only, if an IT method is used with a suitable regression method for LSNMs and a consistent HSIC estimator, then the IT method is consistent for inferring causal direction for LSNMs.*

It is important to note that, while the suitability of regression methods is a valuable property for IT-based methods aiding in the determination of causal directions using independence testing, it does not guarantee a higher likelihood for the causal direction compared to the anti-causal direction under noise misspecification. Therefore, suitability does not offer the same robustness guarantee for ML-based methods.

## K.1    Suitability: empirical results

Table 5: Suitability of the flow estimator $\mathbf{T}$ of CAREFL-M in the causal direction under noise misspecification and misleading CVs. $\mathbf{T}$ is trained with a Laplace prior. The original dataset with size $2N$ is split into two: 50% as training set and 50% as testing set. $\overline{\mathbb{V}}[Y|X] > \overline{\mathbb{V}}[X|Y]$ indicates misleading CVs in the dataset. LSNM definition is in Appendix M.

(a) **LSNM-tanh-exp-cosine** and **continuousBernoulli(0.9)** noise. $\overline{\mathbb{V}}[Y|X]$ vs. $\overline{\mathbb{V}}[X|Y]$: 0.324 vs. 0.291.

|  | N=50 | N=500 | N=1000 | N=5000 |
|---|---|---|---|---|
| $[S_{Z_X}, S_{Z_Y}]$ | [0.02406, 0.01283] | [0.00194, 0.00204] | [0.00094, 0.00092] | [0.0002, 0.00018] |

(b) **LSNM-sine-tanh** and **uniform(−1, 1)** noise. $\overline{\mathbb{V}}[Y|X]$ vs. $\overline{\mathbb{V}}[X|Y]$: 0.422 vs. 0.367.

|  | N=50 | N=500 | N=1000 | N=5000 |
|---|---|---|---|---|
| $[S_{Z_X}, S_{Z_Y}]$ | [0.0081, 0.00285] | [0.00039, 0.00031] | [0.0002, 0.00017] | [0.00003, 0.00003] |

(c) **LSNM-sigmoid-sigmoid** and **exponential(1)** noise. $\overline{\mathbb{V}}[Y|X]$ vs. $\overline{\mathbb{V}}[X|Y]$: 0.927 vs. 0.657.

|  | N=50 | N=500 | N=1000 | N=5000 |
|---|---|---|---|---|
| $[S_{Z_X}, S_{Z_Y}]$ | [0.00979, 0.00443] | [0.00074, 0.00058] | [0.00045, 0.00026] | [0.00005, 0.00005] |

Let *suitability value $S$* be the left-hand side of Equation (9). Table 5 shows an empirical evaluation of $S$, for the flow estimator $\mathbf{T}$ in the causal direction. We generate data from 3 synthetic LSNMs, and evaluate $\mathbf{T}$ under noise misspecification and misleading CVs. We find that as the sample size grows, $S$ approaches 0. In other words, $\mathbf{T}$ *is empirically suitable under noise misspecification and misleading CVs.* Therefore, Proposition K.1 entails that CAREFL-H based on $\mathbf{T}$ and a consistent HSIC estimator is empirically consistent for inferring causal direction in LSNMs under these conditions.

Because $\mathbf{T}$ in CAREFL-M [11] uses neural networks to approximate the observed data distribution, it is difficult to provide a theoretical guarantee of suitability. Therefore, although our experiments indicate that $\mathbf{T}$ is often suitable in practice, we do not claim that it is suitable for all LSNMs. For

example, it may not be suitable in the low-noise regime when the LSNMs are close to deterministic. It is well-known that since neural network models are universal function approximators, we can expect them to fit many regression functions, although it is difficult to prove suitability theoretically. For these reasons, we do not claim a priori that flow models are suitable, but we provide empirical evidence that with careful training and hyperparameter selection, we can expect them to be suitable in practice.

### K.2 Proof outline for Proposition K.1

For data pairs $(X, Y)$ generated by an LSNM model $X \rightarrow Y$, we have:

1. Under suitability, the reconstructed noise $\widehat{Z}_Y$ approaches the ground-truth noise $Z_Y$.

2. By employing a consistent HSIC estimator, the estimated HSIC value $\widehat{HSIC}$ converges to the true HSIC value $HSIC$.

3. In the ground-truth LSNM, $HSIC(X, \widehat{Z}_Y) = 0$ and $HSIC(Y, \widehat{Z}_X) \neq 0$ (i.e. identifiability of LSNMs).

Therefore, $\widehat{HSIC}(X, \widehat{Z}_Y) = 0$ and $\widehat{HSIC}(Y, \widehat{Z}_X) \neq 0$ for LSNMs. We only provide a proof outline here, since a formal proof is analogous to the proof for ANMs in Appendix A of Mooij et al. [14].

## L  Default and alternative hyperparameter values used in Section 7

We use the reported hyperparameter values in CAREFL-M [11] for the Tübingen Cause-Effect Pairs benchmark [14] as the default hyperparameter values in all our experiments:

- The flow estimator $\mathbf{T}$ is parameterized with 4 sub-flows (alternatively: 1, 7 and 10).

- For each sub-flow, $f$, $g$, $h$ and $k$ are modelled as four-layer MLPs with 5 hidden neurons in each layer (alternatively: 2, 10 and 20).

- Prior distribution is Laplace (alternatively: Gaussian prior).

- Adam optimizer [12] is used to train each model for 750 epochs (alternatively: 500, 1000 and 2000).

- L2-penalty strength is 0 by default (alternatively: 0.0001, 0.001, 0.1).

Although we also observe that LOCI-H is more robust than LOCI-M under noise misspecification and misleading CVs, we omit their results because: (1) CAREFL often outperforms LOCI (see Tables 1 and 2). (2) LOCI is fixed as a Gaussian distribution, whereas CAREFL can specify different prior distributions. (3) LOCI uses different set of hyperparameters than CAREFL. So their results cannot be merged into the same figures.

## M  Synthetic SCMs used in Section 7.1

We use the following SCMs to generate synthetic datasets:

$$
\begin{aligned}
\text{LSNM-tanh-exp-cosine:} \quad & Y := \tanh(X \cdot \theta_1) \cdot \theta_2 + e^{\cos(X \cdot \psi_1) \cdot \psi_2} \cdot Z_Y \\
\text{LSNM-sine-tanh:} \quad & Y := \sin(X \cdot \theta_1) \cdot \theta_2 + (\tanh(X \cdot \psi) + \phi) \cdot Z_Y \quad\quad (10) \\
\text{LSNM-sigmoid-sigmoid:} \quad & Y := \sigma(X \cdot \theta_1) \cdot \theta_2 + \sigma(X \cdot \psi_1) \cdot \psi_2 \cdot Z_Y
\end{aligned}
$$

where $Z_Y$ is the ground-truth noise sampled from one of the following distributions: $continuousBernoulli(0.9)$, $uniform(-1, 1)$, $exponential(1)$, $beta(0.5, 0.5)$, $Gaussian(0, 1)$ and $Laplace(0, 1)$. $\sigma$ is the sigmoid function. Although we did not prove identifiability with $beta(0.5, 0.5)$ noise in Theorem C.2, empirically we find it is identifiable. Following [33, 34], each $\theta$ and $\psi$ are sampled uniformly from range $[-2, -0.5] \cup [0.5, 2]$. $\phi$ is sampled uniformly from range $[1, 2]$ to make the $tanh$ function positive. The number of data pairs in each synthetic dataset is $N \in \{500, 5000\}$.

## N Hyperparameter values of CAREFL-H for Table 2b

Following Khemakhem et al. [11], we use a single set of hyperparameters for all 99 datasets, which is found by grid search. To acquire the result of CAREFL-H in Table 2b, the hyperparameter values used are as follows:

- Number of hidden neurons in each layer of the MLPs: 2
- Number of sub-flows: 10
- Training dataset = testing dataset = 100%.

The rest of the hyperparameter values are identical to the default ones.

## O Running time

The running time of CAREFL-H is slightly longer than CAREFL-M, due to the additional independence tests, i.e. $HSIC(X_{test}, \widehat{Z}_Y)$ and $HSIC(Y_{test}, \widehat{Z}_X)$. Please see Table 6. The running time is measured on a computer running Ubuntu 20.04.5 LTS with Intel Core i7-6850K 3.60GHz CPU and 32 GB memory. No GPUs are used.

Table 6: **Accuracy running time in seconds** over 10 datasets generated by SCM **LSNM-sine-tanh** (definition in Appendix M) with $N = 10,000$ samples. $\alpha = 1$. Ground-truth noise distribution is $uniform(-1, 1)$. Model prior distribution is $Gaussian(0, 1)$.

| Method | Average Running Time in Seconds |
|---------|---------------------------------|
| CAREFL-M | 33.23 |
| CAREFL-H | 39.38 |

## P Additive noise models under misleading conditional variances

We show the accuracy of ANM ML methods with ANM data under misleading CVs, and compare with LSNM ML methods. The ANM data is generated by the following SCMs:

$$
\begin{aligned}
\text{ANM-sine:} \quad & Y := \sin(X \cdot \theta_1) \cdot \theta_2 + Z_Y \\
\text{ANM-tanh:} \quad & Y := \tanh(X \cdot \theta_1) \cdot \theta_2 + Z_Y \\
\text{ANM-sigmoid:} \quad & Y := \sigma(X \cdot \theta_1) \cdot \theta_2 + Z_Y
\end{aligned}
\tag{11}
$$

where $Z_Y$ is the ground-truth noise sampled from one of the following distributions: $uniform(-1, 1)$ and $Gaussian(0, 1)$. $\sigma$ is the sigmoid function. Following [33, 34], each $\theta$ is sampled uniformly from range $[-2, -0.5] \cup [0.5, 2]$.

CAREFL-ANM-M is the ANM counterpart of CAREFL-M (see Appendix I). The LSNM flows (see Equation (6)) in CAREFL-M are changed to model ANMs:

$$
\begin{aligned}
X &= t_1 + Z_X \\
Y &= t_2(X) + Z_Y
\end{aligned}
$$

Similar to LSNM ML methods, the accuracy of ANM ML methods is also negatively related to misleading CVs, as shown in Figure 5. Unlike LSNM ML methods, which suffer from misleading CVs under noise misspecification (see Table 1 and Figures 2 and 4), an ANM ML method suffers from misleading CVs with either noise misspecification or correct specification, as illustrated in Figure 5. Figure 1 summarizes the differences between ANM and LSNM ML methods.

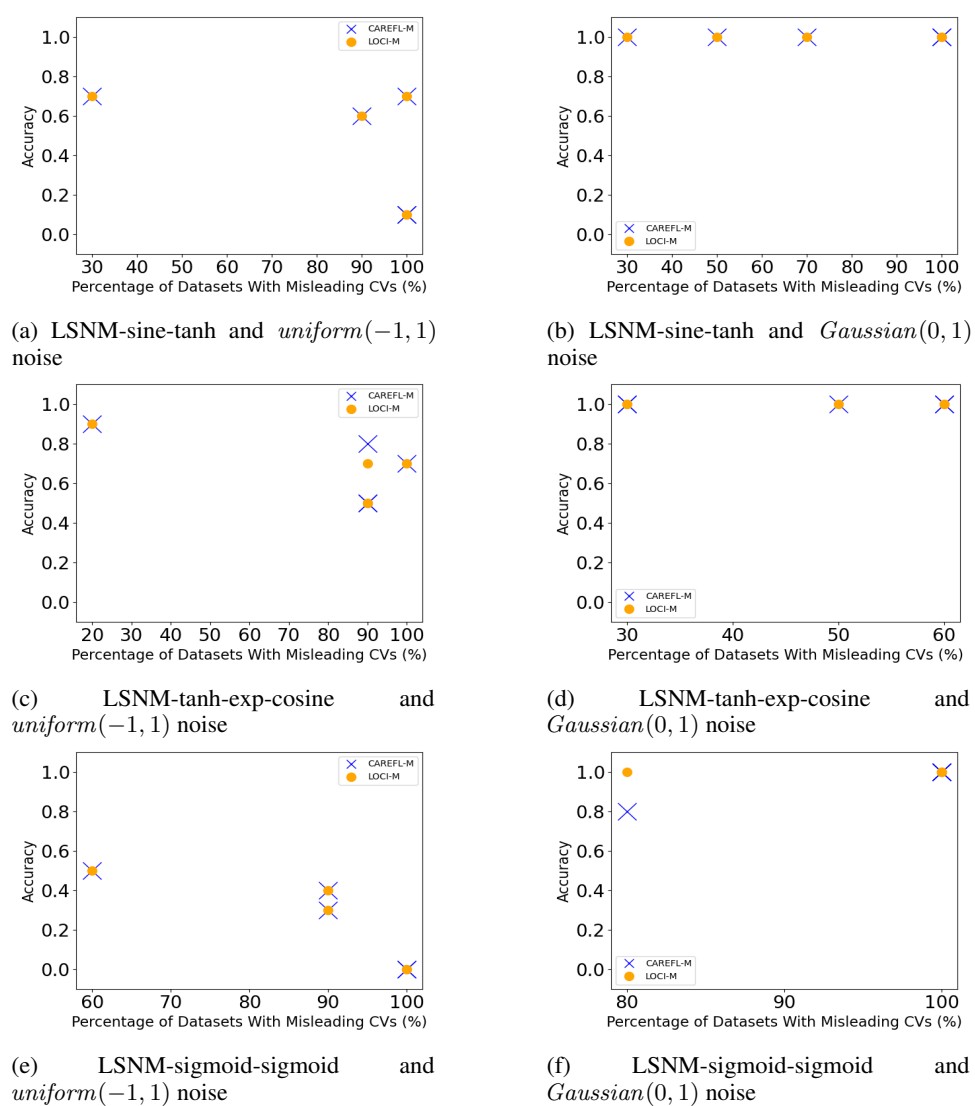

(a) LSNM-sine-tanh and $uniform(-1, 1)$ noise

(b) LSNM-sine-tanh and $Gaussian(0, 1)$ noise

(c) LSNM-tanh-exp-cosine and $uniform(-1, 1)$ noise

(d) LSNM-tanh-exp-cosine and $Gaussian(0, 1)$ noise

(e) LSNM-sigmoid-sigmoid and $uniform(-1, 1)$ noise

(f) LSNM-sigmoid-sigmoid and $Gaussian(0, 1)$ noise

Figure 4: We rewrite Equations (10) as $Y := f(X) + \alpha \cdot g(X) \cdot Z_Y$. Altering $\alpha$ changes the variance of noise, and creates datasets with difference CVs. 10 datasets are generated with each $\alpha$. CVs are computed by binning the true cause. All the methods in the figure assume $Gaussian(0, 1)$ whereas the ground-truth noise is either $uniform(-1, 1)$ or $Gaussian(0, 1)$. All the datasets are standardized to have mean 0 and variance 1.

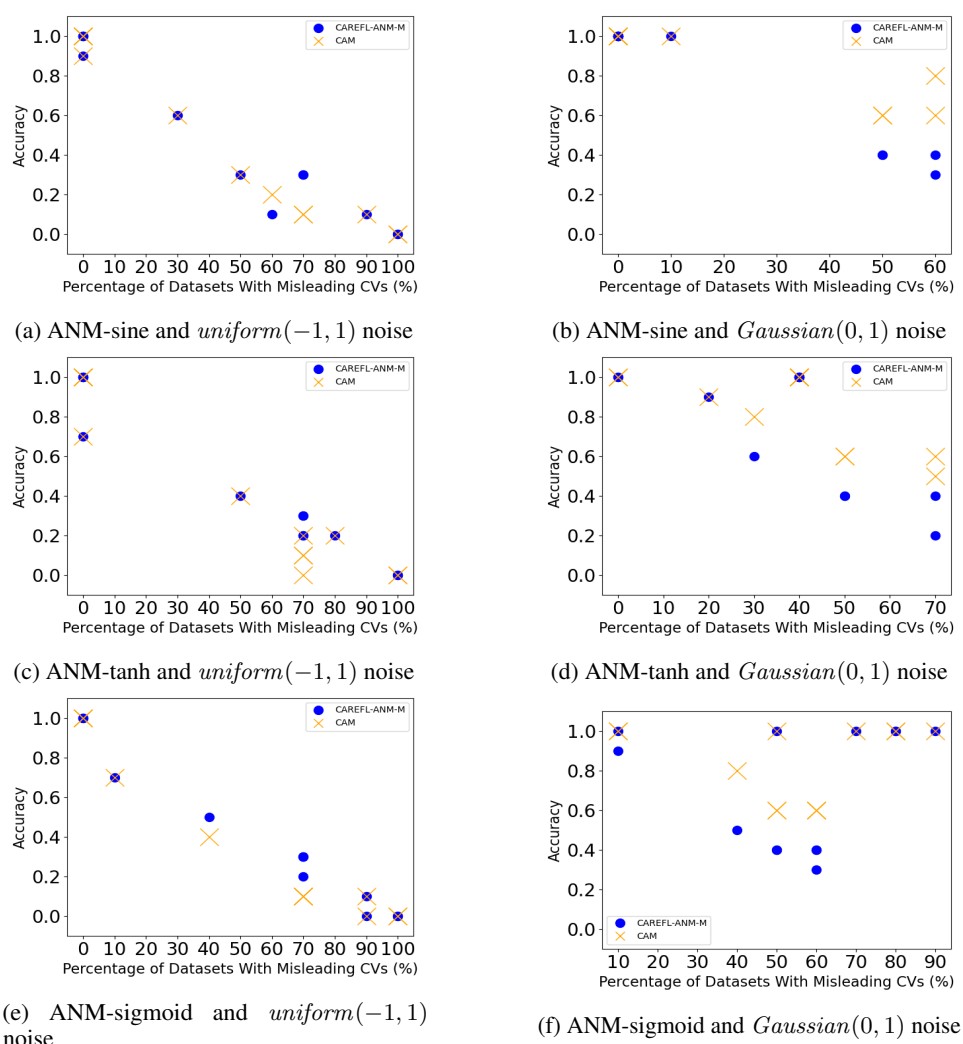

(a) ANM-sine and $uniform(-1, 1)$ noise

(b) ANM-sine and $Gaussian(0, 1)$ noise

(c) ANM-tanh and $uniform(-1, 1)$ noise

(d) ANM-tanh and $Gaussian(0, 1)$ noise

(e) ANM-sigmoid and $uniform(-1, 1)$ noise

(f) ANM-sigmoid and $Gaussian(0, 1)$ noise

Figure 5: We rewrite Equations (11) as $Y := f(X) + \alpha \cdot Z_Y$. Altering $\alpha$ changes the variance of noise, and creates datasets with difference CVs. 10 datasets are generated with each $\alpha$. CVs are computed by binning the true cause. All the methods in the figure assume $Gaussian(0, 1)$ whereas the ground-truth noise is either $uniform(-1, 1)$ or $Gaussian(0, 1)$. All the datasets are standardized to have mean 0 and variance 1.

Table 7: (Best viewed in color) Summary of datasets. Settings in Section 7.1 are color-coded. (1) Blue: with noise misspecification; (2) Red: with more than 50% of datasets having misleading CVs; (3) Brown: both (1) and (2). The datasets in Section 7.1 are generated by LSNMs and the datasets in Sections 7.2 and 7.3 are not.

| Type | Name | Noise | Number of Datasets | Percentage of Datasets With Misleading CVs (%) |
|---|---|---|---|---|
| Synthetic (Section 7.1) | LSNM-tanh-exp-cosine | $Uniform(-1, 1)$ | 10 | 60 |
| | | $Beta(0.5, 0.5)$ | 10 | 80 |
| | | $ContinuousBernoulli(0.9)$ | 10 | 70 |
| | | $Exponential(1)$ | 10 | 20 |
| | | $Gaussian(0, 1)$ | 10 | 40 |
| | | $Laplace(0, 1)$ | 10 | 40 |
| | LSNM-sine-tanh | $Uniform(-1, 1)$ | 10 | 100 |
| | | $Beta(0.5, 0.5)$ | 10 | 100 |
| | | $ContinuousBernoulli(0.9)$ | 10 | 60 |
| | | $Exponential(1)$ | 10 | 20 |
| | | $Gaussian(0, 1)$ | 10 | 60 |
| | | $Laplace(0, 1)$ | 10 | 80 |
| | LSNM-sigmoid-sigmoid | $Uniform(-1, 1)$ | 10 | 90 |
| | | $Beta(0.5, 0.5)$ | 10 | 100 |
| | | $ContinuousBernoulli(0.9)$ | 10 | 80 |
| | | $Exponential(1)$ | 10 | 70 |
| | | $Gaussian(0, 1)$ | 10 | 90 |
| | | $Laplace(0, 1)$ | 10 | 100 |
| Synthetic (Section 7.2) | SIM | N/A | 100 | 40 |
| | SIM-c | | 100 | 46 |
| | SIM-ln | | 100 | 23 |
| | SIM-G | | 100 | 29 |
| Real-World (Section 7.3) | Tübingen Cause-Effect Pairs | N/A | 99 | 40 |

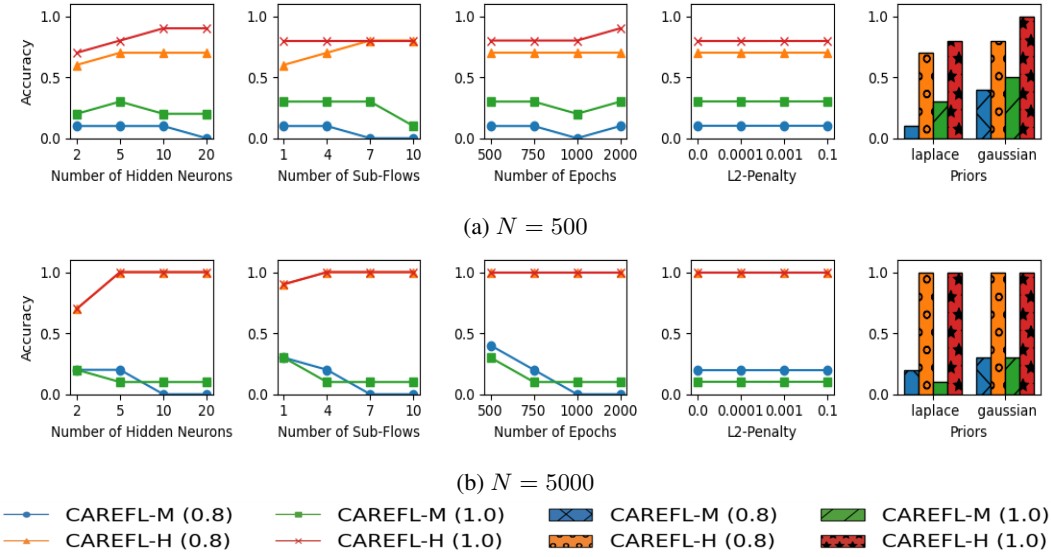

(a) $N = 500$

(b) $N = 5000$

CAREFL-M (0.8)    CAREFL-M (1.0)    CAREFL-M (0.8)    CAREFL-M (1.0)
CAREFL-H (0.8)    CAREFL-H (1.0)    CAREFL-H (0.8)    CAREFL-H (1.0)

Figure 6: **Accuracy** over 10 datasets: **LSNM-tanh-exp-cosine** and **uniform**$(-1, 1)$ noise.

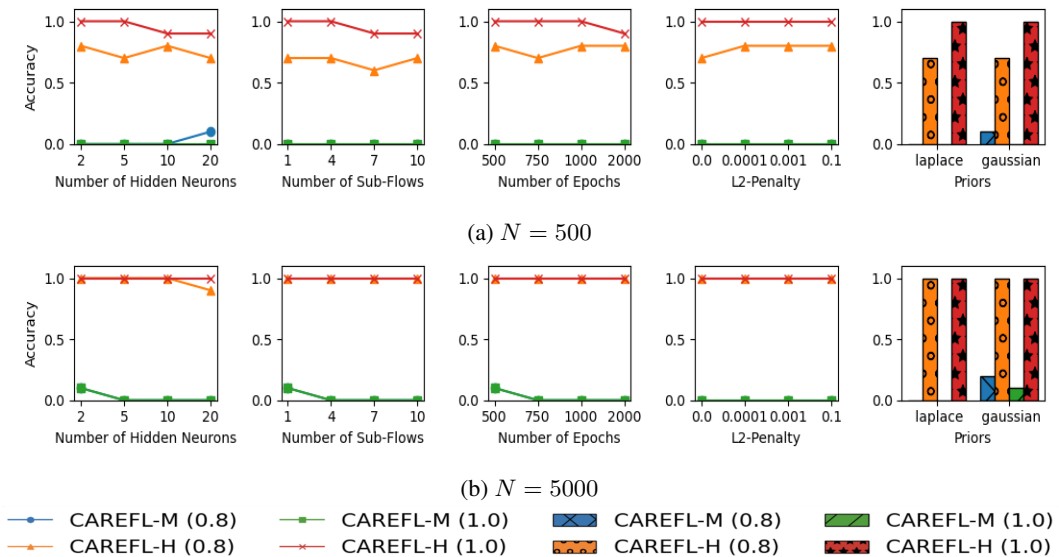

(a) $N = 500$

(b) $N = 5000$

Figure 7: **Accuracy** over 10 datasets: **LSNM-tanh-exp-cosine** and $\mathbf{beta(0.5, 0.5)}$ noise.

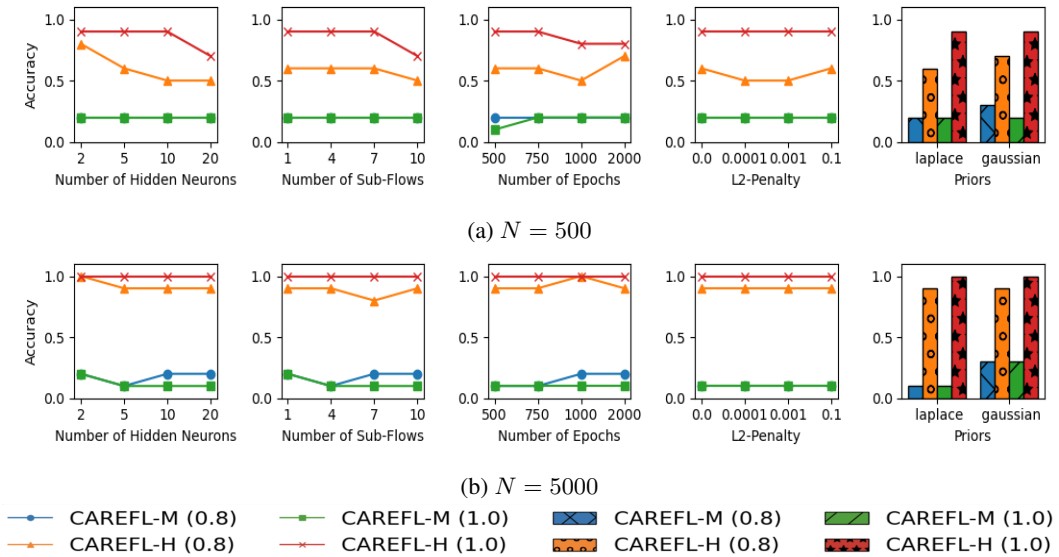

(a) $N = 500$

(b) $N = 5000$

Figure 8: **Accuracy** over 10 datasets: **LSNM-tanh-exp-cosine** and $\mathbf{continuousBernoulli(0.9)}$ noise.

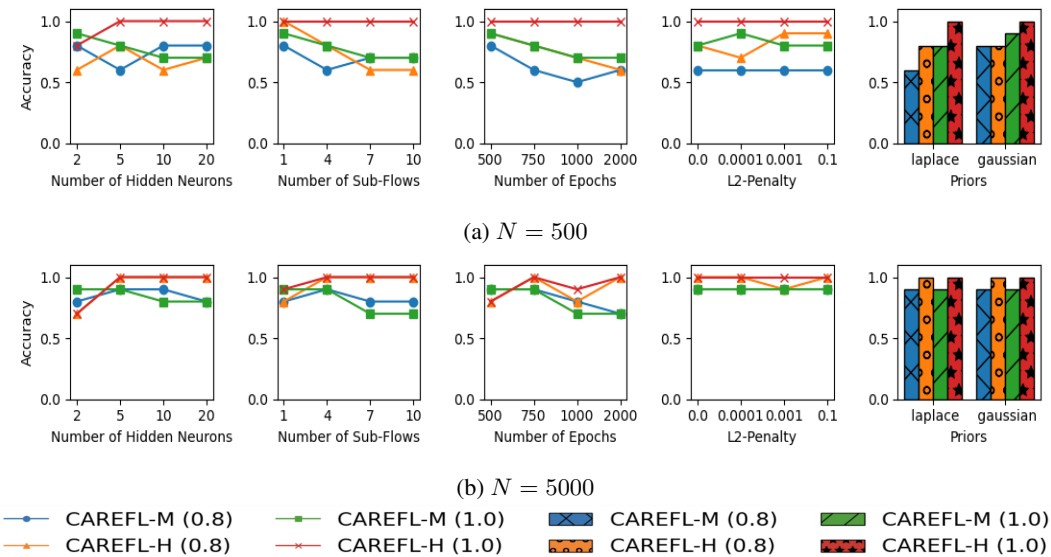

Figure 9: **Accuracy** over 10 datasets: **LSNM-tanh-exp-cosine** and **exponential**$(\mathbf{1})$ noise.

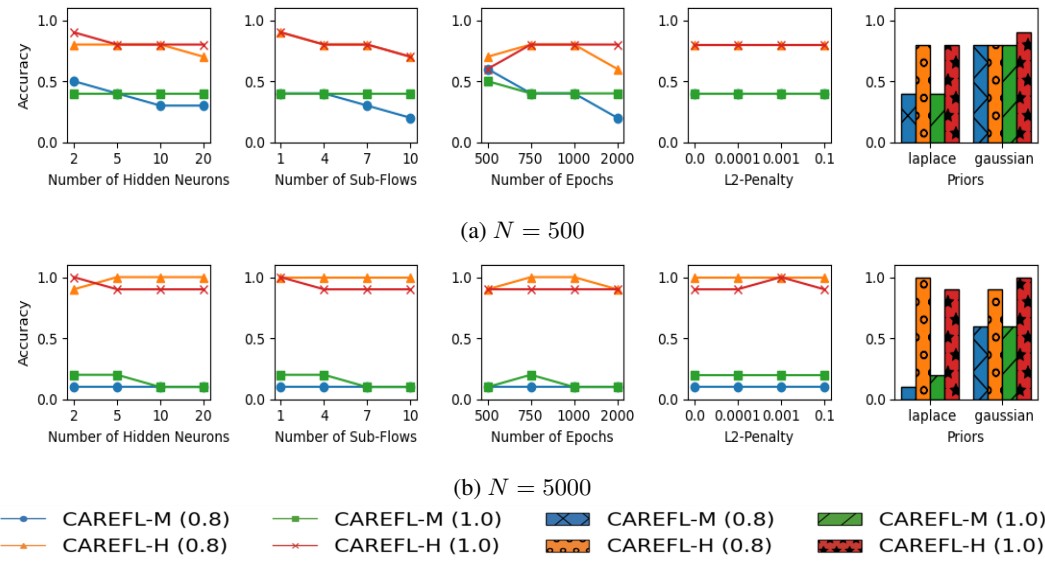

Figure 10: **Accuracy** over 10 datasets: **LSNM-sine-tanh** and **uniform**$(-\mathbf{1}, \mathbf{1})$ noise.

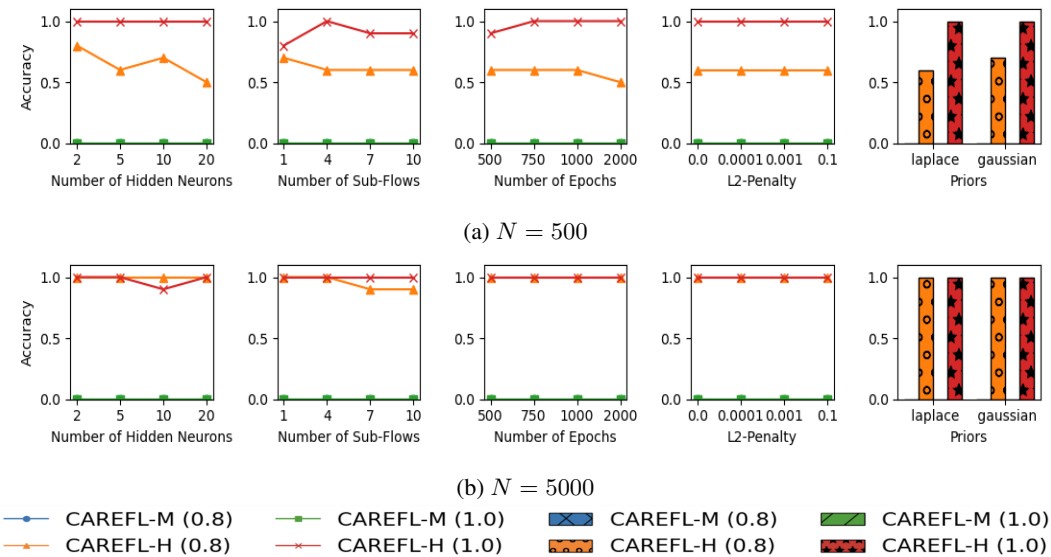

Figure 11: **Accuracy** over 10 datasets: **LSNM-sine-tanh** and $\mathbf{beta(0.5, 0.5)}$ noise.

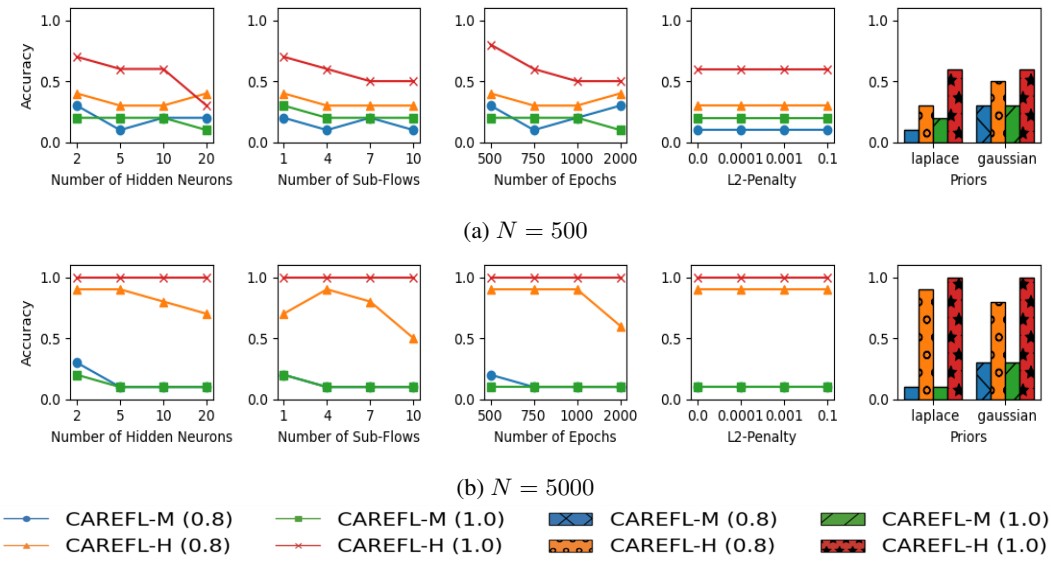

Figure 12: **Accuracy** over 10 datasets: **LSNM-sine-tanh** and **continuousBernoulli(0.9)** noise.

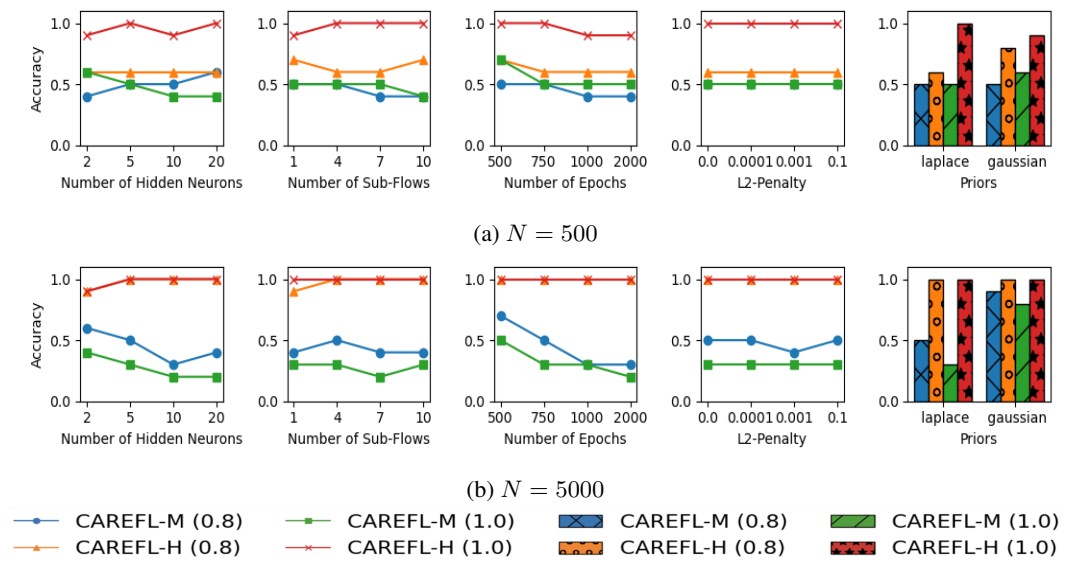

Figure 13: **Accuracy** over 10 datasets: **LSNM-sine-tanh** and **exponential(1)** noise.

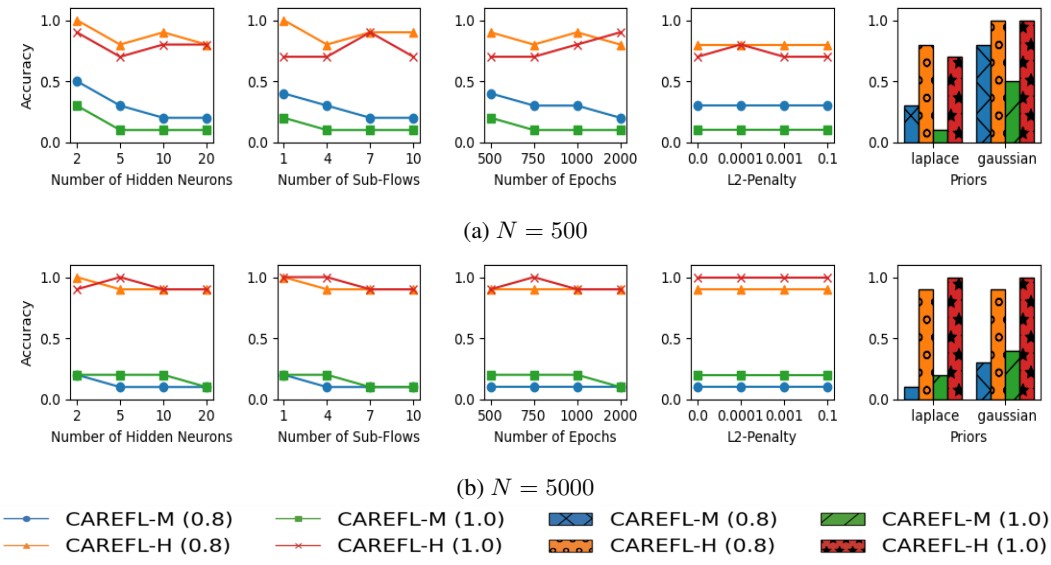

Figure 14: **Accuracy** over 10 datasets: **LSNM-sigmoid-sigmoid** and **uniform(−1, 1)** noise.

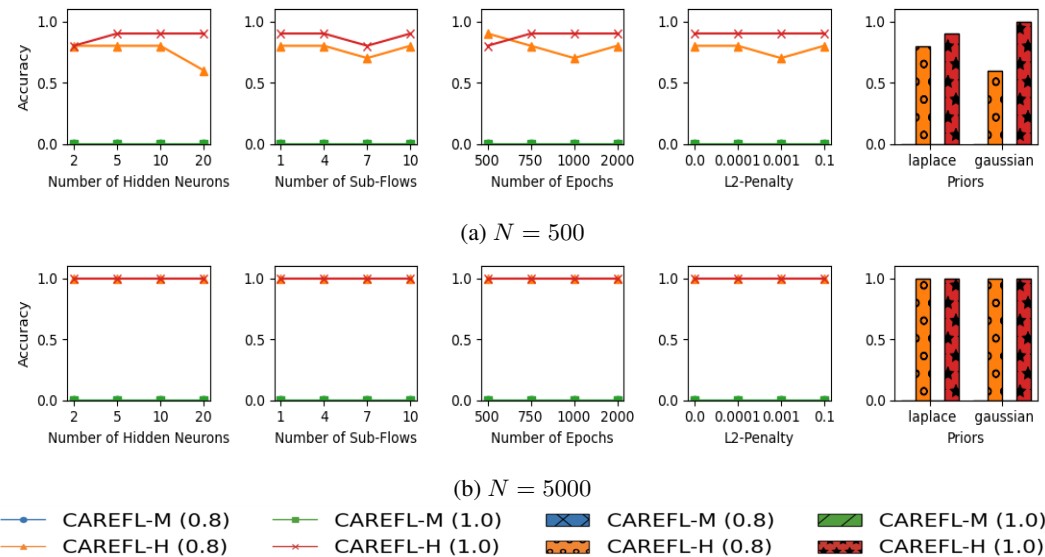

Figure 15: **Accuracy** over 10 datasets: **LSNM-sigmoid-sigmoid** and $\mathbf{beta(0.5, 0.5)}$ noise.

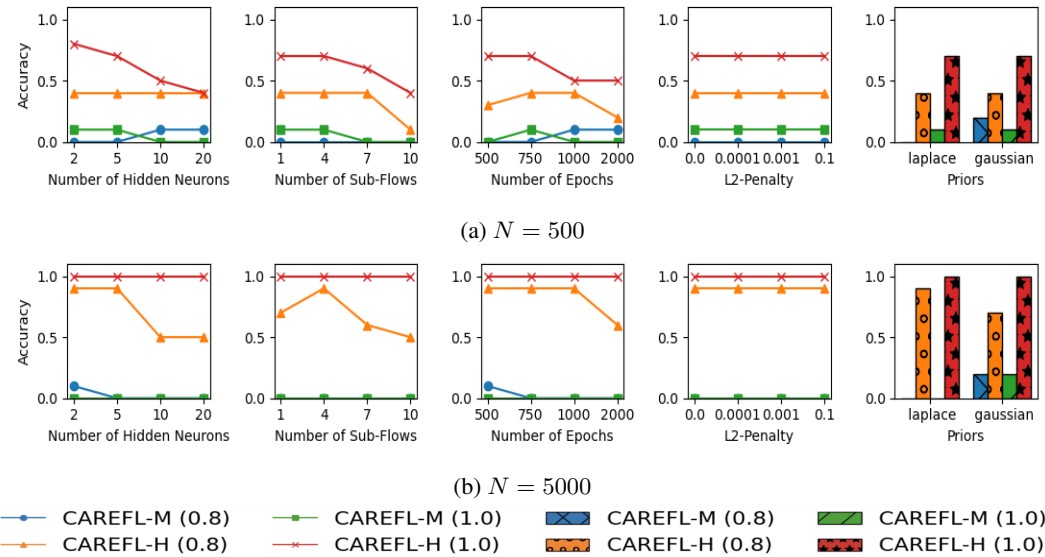

Figure 16: **Accuracy** over 10 datasets: **LSNM-sigmoid-sigmoid** and $\mathbf{continuousBernoulli(0.9)}$ noise.

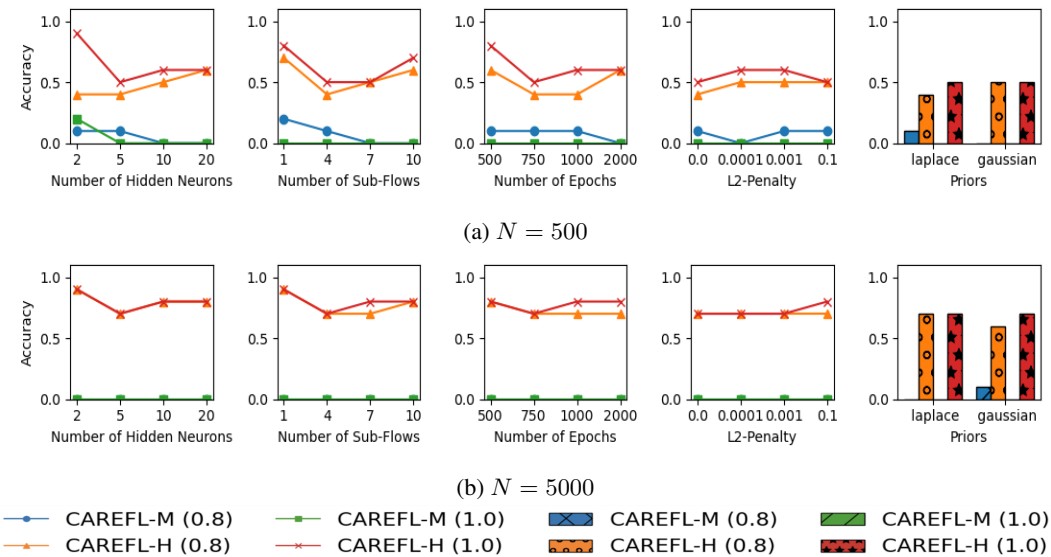

(a) $N = 500$

(b) $N = 5000$

| CAREFL-M (0.8) | CAREFL-M (1.0) | CAREFL-M (0.8) | CAREFL-M (1.0) |
| CAREFL-H (0.8) | CAREFL-H (1.0) | CAREFL-H (0.8) | CAREFL-H (1.0) |

Figure 17: **Accuracy** over 10 datasets: **LSNM-sigmoid-sigmoid** and **exponential(1)** noise.

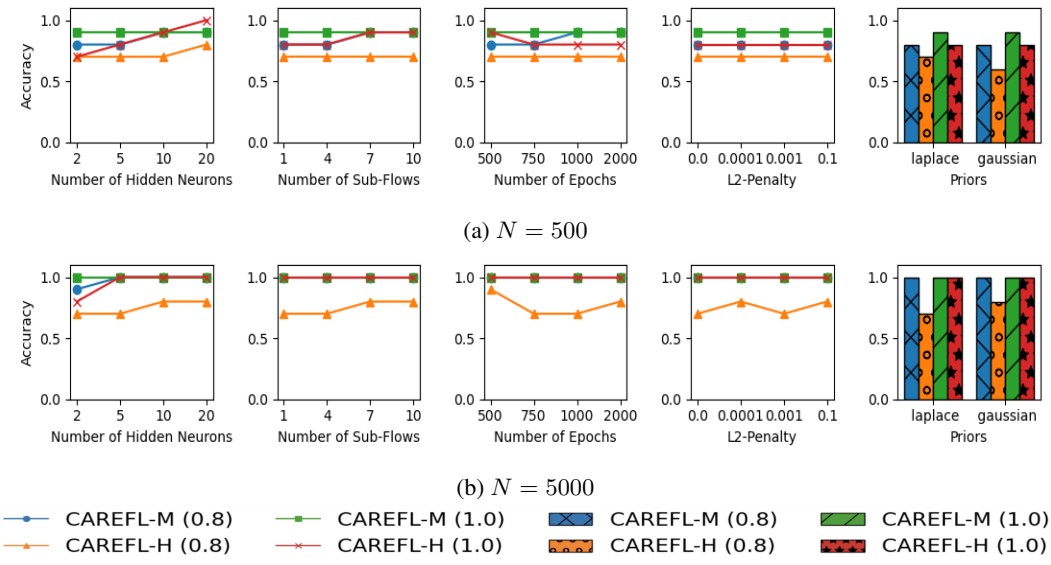

(a) $N = 500$

(b) $N = 5000$

| CAREFL-M (0.8) | CAREFL-M (1.0) | CAREFL-M (0.8) | CAREFL-M (1.0) |
| CAREFL-H (0.8) | CAREFL-H (1.0) | CAREFL-H (0.8) | CAREFL-H (1.0) |

Figure 18: **Accuracy** over 10 datasets: **LSNM-tanh-exp-cosine** and **Gaussian(0, 1)** noise.

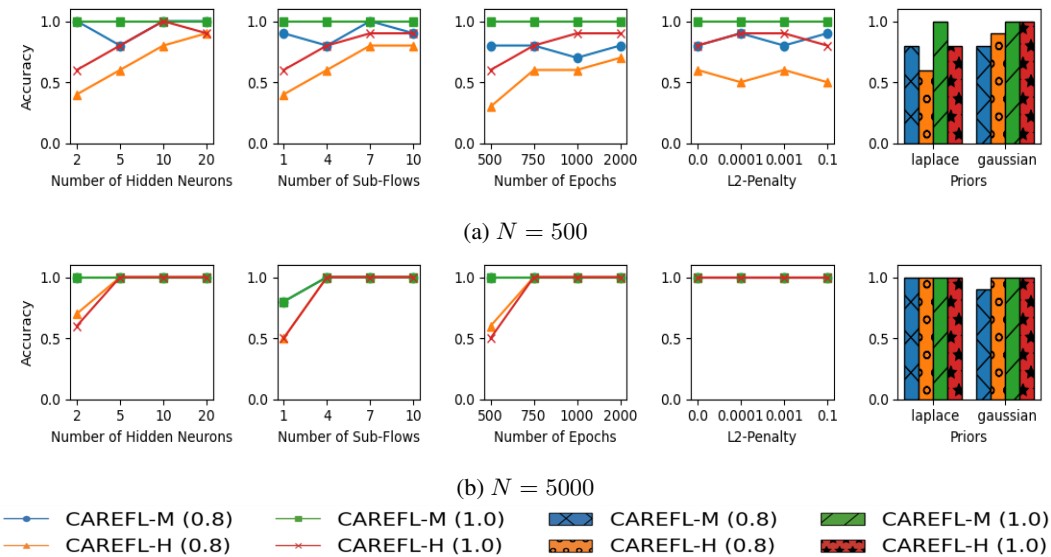

Figure 19: **Accuracy** over 10 datasets: **LSNM-tanh-exp-cosine** and $\mathbf{Laplace(0, 1)}$ noise.

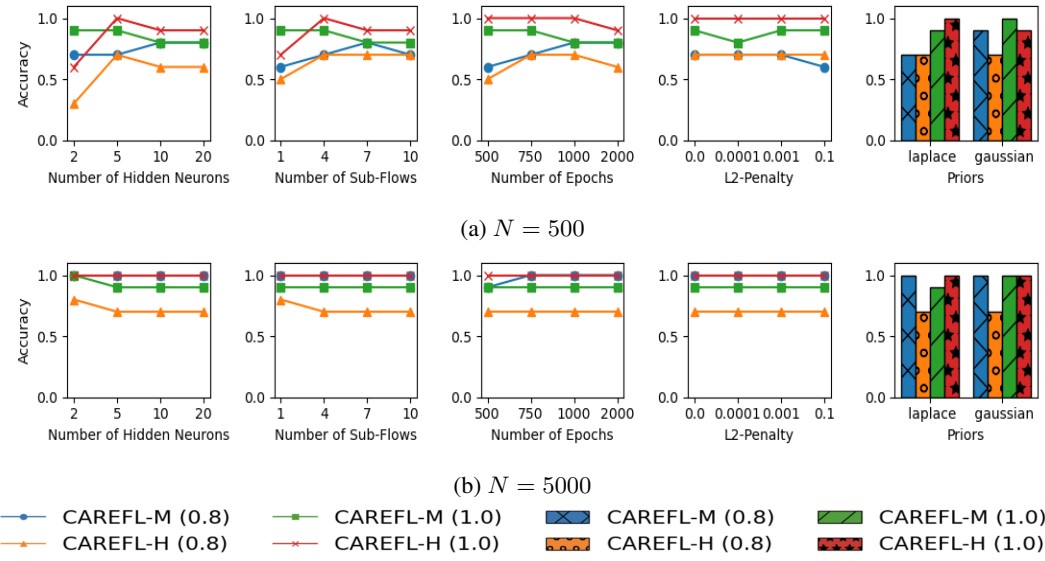

Figure 20: **Accuracy** over 10 datasets: **LSNM-sine-tanh** and $\mathbf{Gaussian(0, 1)}$ noise.

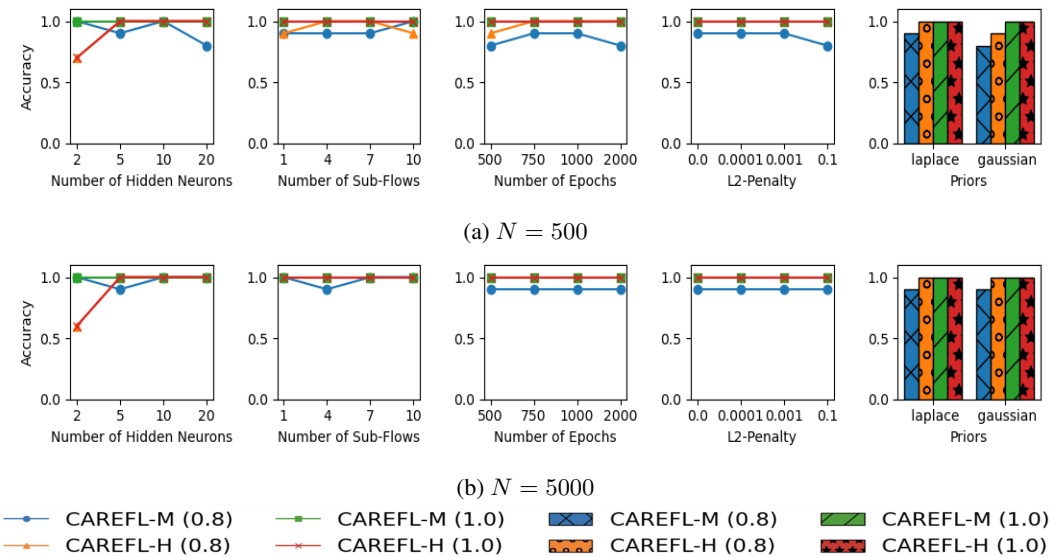

(a) $N = 500$

(b) $N = 5000$

Figure 21: **Accuracy** over 10 datasets: **LSNM-sine-tanh** and **Laplace**$(0, 1)$ noise.

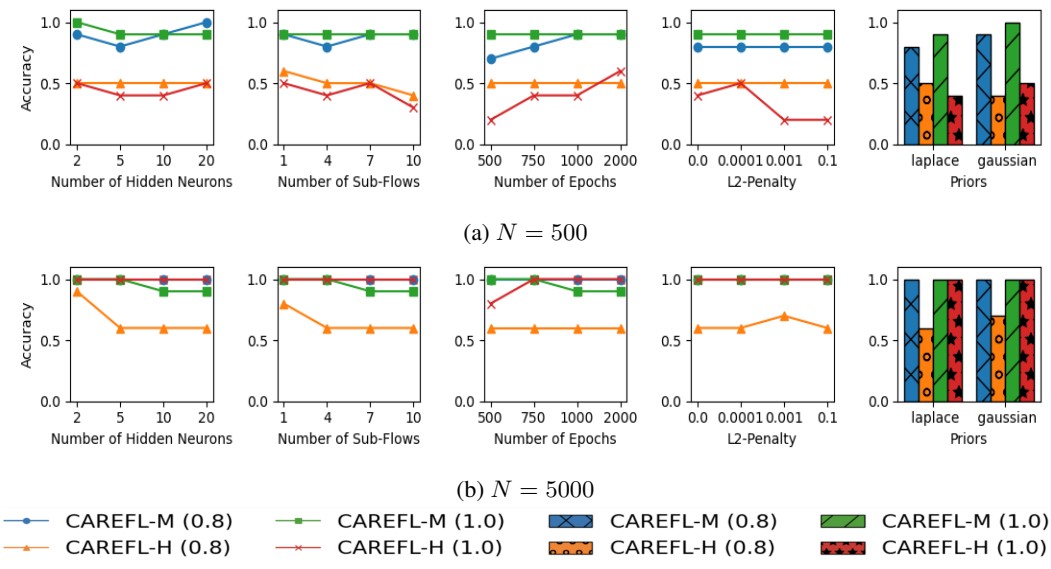

(a) $N = 500$

(b) $N = 5000$

Figure 22: **Accuracy** over 10 datasets: **LSNM-sigmoid-sigmoid** and **Gaussian**$(0, 1)$ noise.

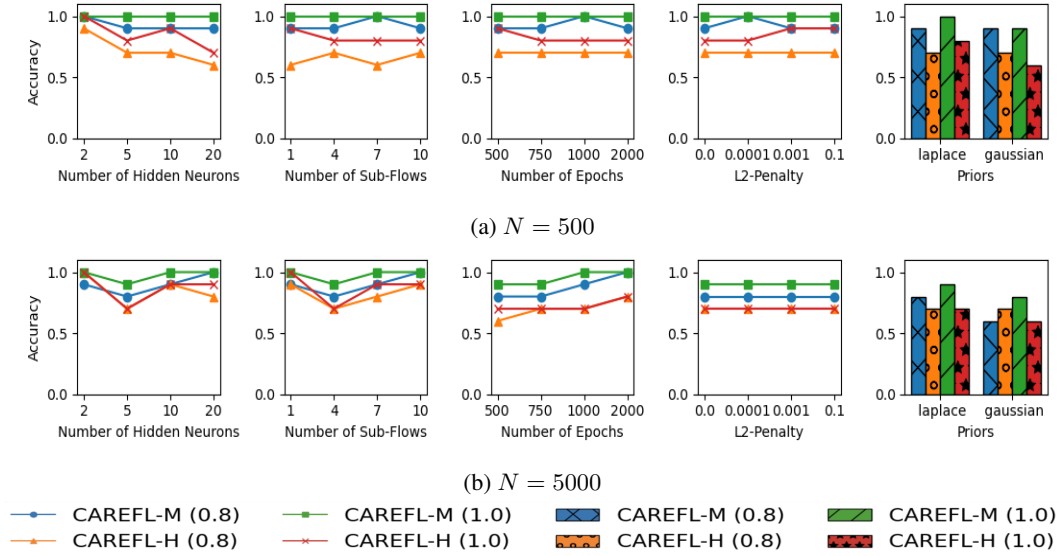

Figure 23: **Accuracy** over 10 datasets: **LSNM-sigmoid-sigmoid** and **Laplace(0, 1)** noise.

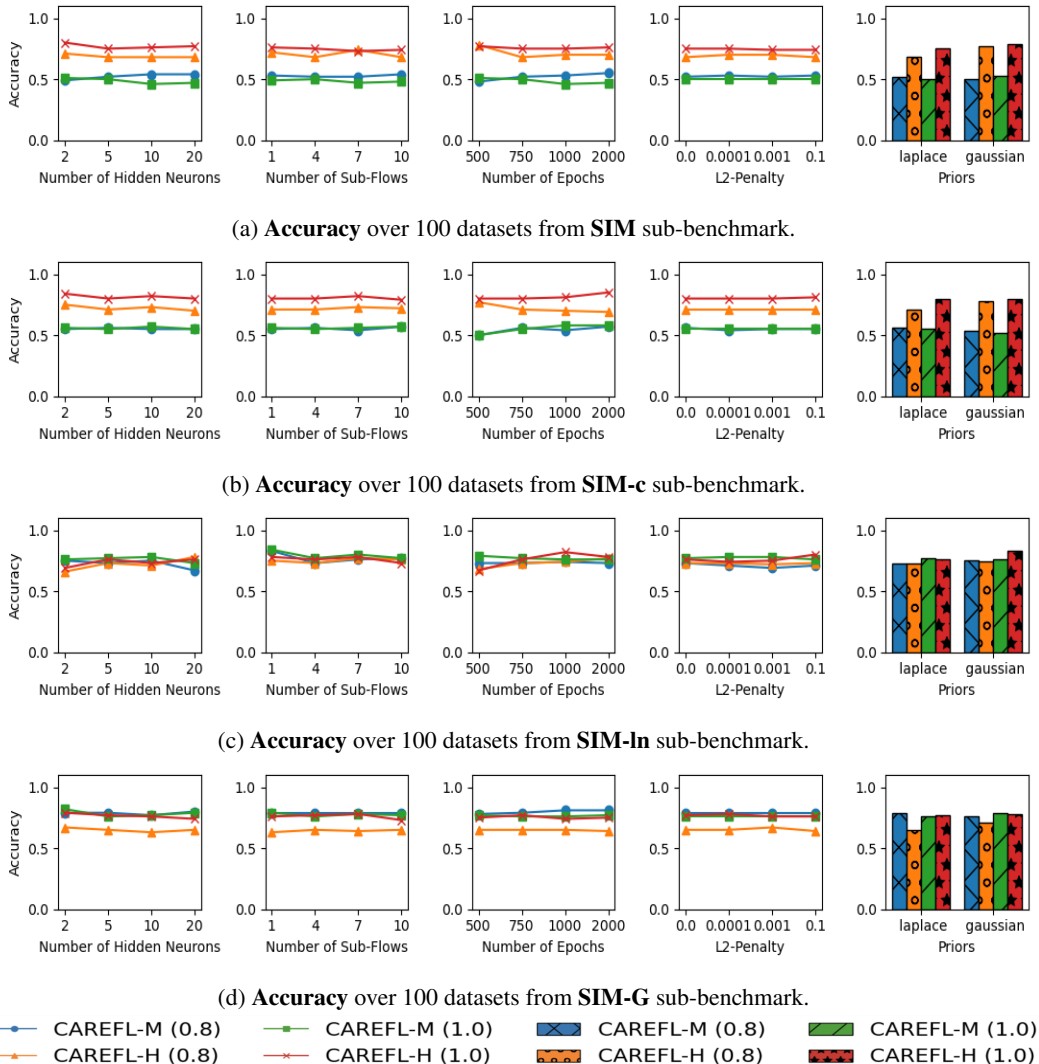

(a) **Accuracy** over 100 datasets from **SIM** sub-benchmark.

(b) **Accuracy** over 100 datasets from **SIM-c** sub-benchmark.

(c) **Accuracy** over 100 datasets from **SIM-ln** sub-benchmark.

(d) **Accuracy** over 100 datasets from **SIM-G** sub-benchmark.

| | | | |
|---|---|---|---|
| CAREFL-M (0.8) | CAREFL-M (1.0) | CAREFL-M (0.8) | CAREFL-M (1.0) |
| CAREFL-H (0.8) | CAREFL-H (1.0) | CAREFL-H (0.8) | CAREFL-H (1.0) |

Figure 24: Results with **SIM benchmarks**.

