# OpenReview forum: "Cause-Effect Inference in Location-Scale Noise Models: Maximum Likelihood vs. Independence Testing"
_NeurIPS.cc/2023/Conference — NeurIPS 2023 poster_

### Official Review · Reviewer_ttx5 · 2023-07-03

**Soundness:** 3 good
**Presentation:** 4 excellent
**Contribution:** 3 good
**Rating:** 6
**Confidence:** 4

**Summary:**

The paper discusses the sensitivity of the recently introduced heteroscedastic noise model to model specification for causal discovery in a bivariate scenario. Specifically, it focuses on the impact of misspecifying the assumed noise distribution, which leads to poor performance when choosing the direction based on maximizing the likelihood. Instead, the paper proposes the use of independence testing of the residuals and presents an improved model. The newly proposed model demonstrates promising results in both real-world and synthetic datasets.

**Strengths:**

* Excellent and clear introduction.
* Comprehensive literature review with useful categorization of different approaches.
* Highlights flaws in the existing prior approach.
* Extensive experiments using popular causal discovery datasets.

Also see questions section.

**Weaknesses:**

* The novelty appears to be limited: The result seems rather incremental, considering that the implications of model misspecification aren't surprising, and the use of independence tests is a relatively common way to utilize functional causal models for causal discovery.
* There is a limited discussion about the work "Identifying Patient-Specific Root Causes with the Heteroscedastic Noise Model" by Strobl et al., given that they don't rely on maximizing the likelihood for fitting the model, do not seem to depend on prior knowledge about the noise distribution and also focus on causal discovery.

Also see questions section.

**Questions:**

The paper is generally well-written and includes an in-depth comparison with related work. Some general remarks and questions:

* You mention LiNGAM as an independence testing approach. However, strictly speaking, it only relies on the ICA components (assuming non-Gaussian noise), which is not utilizing independence tests.
* The reference for GRCI appears to be incorrect.
* There needs to be a more comprehensive comparison with the work by Strobl et al., as they do not rely on prior knowledge about the noise or exploitation of the conditional variance, which is the main critique of this paper.
* The meaning of Figure 1 could be clearer. For instance, what does “high” and “low” exactly mean for accuracy?
* When considering the conditional variance, it seems evident that this is primarily a matter of scaling. One can simply scale X and Y differently to arbitrarily flip edges for approaches that do not carefully scale the variables as part of their methodology.
* Table 1 seems slightly out of place and appears to be more appropriate for the experiment section.
* In the case of CAREFL-M, isn't the prior distribution merely a convention, since they learn a transformation function of the noise, i.e., they use g(N) and not N directly. Therefore, the Gaussian noise can be transformed into 'arbitrary' noise distributions. Perhaps, I'm missing something here?

--Update after rebuttal-- I have read the rebuttal and further discussed with the authors. Many of my concerns have been addressed and I raised the score by 1.

**Limitations:**

No concerns regarding societal impact. Regarding technical limitations, this paper focuses primarily on improving an existing methodology and, thus, shares the same limitations as the original work. However, it aims to make the methodology less sensitive to model misspecifications.

---

> ### Author Rebuttal · Authors · 2023-08-07
>
> ### Re: Weakness, Novelty & Limitations
> Please refer to the 2nd global response (*Re: Contribution*) regarding the novelty / contribution of the work.
>
> ### Re: Weakness, Little Discussion of GRCI
> We introduced GRCI in our paper on line 90 and conducted empirical comparisons with it in the experiment section. The primary focus of our work is addressing the challenges of noise misspecification and misleading CVs in LSNMs. It's important to note that the authors of GRCI do not discuss this problem in their work.
>
> We used GRCI as a comparison baseline for our proposed algorithm CAREFL-H. While both GRCI and CAREFL-H achieve similar accuracy against the Tübingen Cause-Effect Pairs benchmark, CAREFL-H demonstrates superior performance over GRCI in all settings when evaluated against the popular SIM benchmark (Table 3a).
>
> ### Re: Questions, LiNGAM
> We would like to clarify that in our paper, we are referring to the Direct-LiNGAM algorithm [25], not the ICA-LiNGAM [24]. Direct-LiNGAM is an independence-based algorithm.
>
> As an additional clarification, the ICA-LiNGAM algorithm is also related to independence, specifically focusing on the mutual independence among noise terms in the context of SCMs. This concept is equivalent to mutual independence among the sources in the context of ICA.
>
> ### Re: Questions, More Comparison with GRCI
> Our paper is primarily centered around identifying the challenges faced by likelihood-based methods in LSNMs, which is the main focus of our research. On the other hand, GRCI is not a likelihood-based method, and it lies outside the primary scope of our paper. However, for comparison purposes, we empirically evaluate and compare with GRCI since both GRCI and our proposed CAREFL-H are independence-based approaches.
>
> Although both our work and GRCI involve causal discovery in LSNMs, the main focus of the two papers differs significantly. Our primary emphasis is addressing the specific issue of likelihood-based methods in LSNMs. The authors of GRCI do not discuss this issue in their work. In contrast, GRCI's primary focus is on detecting the root causes of events, such as causes of diseases. These distinctions underscore the unique contributions and research objectives of the two papers.
>
> ### Re: Questions, What does "High" and "Low" mean in Figure 1.
> The primary purpose of Figure 1 is to provide a qualitative summary of the quantitative results from Tables 1 & 5 and Figures 4 & 5. The labels "high" and "low" are used to convey specific meanings:
>
> - When we refer to "high" accuracy, we imply superior accuracy that is supported by a robust theoretical explanation, such as Corollary 4.3 or the identifiability results.
> - On the other hand, for "low" accuracy, we are indicating a decrease in accuracy after switching settings, for example, transitioning from LSNM to ANM, or from having no misleading CVs to having misleading CVs. This inferior accuracy is also well-grounded by theoretical explanations, such as Corollary 4.3 and Proposition H.1.
>
> It's important to note that these "high" and "low" accuracy assessments are demonstrated quantitatively in Tables 1 & 5 and Figures 4 & 5. For instance, the "high" accuracy in the path of (Noise misspecification "No" -> Misleading CVs "Yes" -> LSNM) in Figure 1 corresponds to a value of 1.0 in Table 1, while the "low" accuracy in the path of (Noise misspecification "Yes" -> Misleading CVs "Yes" -> LSNM) in Figure 1 corresponds to a value of 0.1 in Table 1.
>
> ### Re: Questions, Scaling
> We would like to answer the question both theoretically and empirically:
>
> - In **ANMs**, scaling may **lead to a flip in likelihoods**. However, our Lemma 4.1 provides a crucial finding that scaling **does not result in flipped likelihoods** in **LSNMs** (under correct noise specification). This discovery highlights a substantial difference between LSNMs and ANMs, as elaborated on lines 159-161 and Appendix O.
> - To address the issue of scaling, a common approach is to standardize variables to have mean 0 and variance 1. We have adopted this standardization for all our experiments, as indicated on lines 48-49, 223, and Table 1. Consequently, the identified problem persists in LSNMs even when X and Y are standardized to the same scale.
>
> ### Re: Questions, Isn't Gaussian Distribution Merely a Convention
> In many machine learning tasks, using a Gaussian distribution as the model prior is a common convention, and this Gaussian prior can be transformed into arbitrary distributions (e.g. in VAEs or Normalizing Flows).
>
> However, causal discovery relies on specific assumptions to determine the correct causal direction. Without these assumptions, a model becomes too flexible, making it impossible to distinguish between X->Y and X<-Y. The identifiability theorems of LSNMs for ML model selection necessitate the correct prior distribution $N$ to distinguish the directions.
>
> Please note that the LSNM model has $g(X)$ not $g(N)$ as you write. This may just be a typo, but it is important because the $g(X)$ function does not directly transform the noise $N$. It simply rescales it through the transformation $g(X) \cdot N$. The $g(X) \cdot N$ transformation is not powerful enough to "correct" a misspecified prior on $N$. If we consider $g(X) \cdot N$ as the "noise", the LSNM SCM becomes $Y= f(X) + N'$, where $N' = g(X) \cdot N$. This new SCM may appear similar to an ANM, but it is not, as $N'$ and $X$ are not independent. As of now, we are not aware of any identifiable SCMs in this particular form.
>
> ### Re: Questions, Reference For GRCI is Incorrect.
> The GRCI paper was initially available on arXiv and accepted by the Journal of Computational Science on 14 June 2023, which is after the NeurIPS submission deadline. We have updated the reference in the paper to reflect this change.
>
> ### Re: Questions, Table 1 is More Appropriate for the Experiment Section.
> We utilize Table 1 as a preview to captivate readers' interest and motivate the problem we are addressing in our work.

---

> > ### Comment · Reviewer_ttx5 · 2023-08-11
> >
> > I want to thank the authors for their thoughtful response to my raised concerns.
> >
> > > Re: Weakness, Little Discussion of GRCI
> >
> > I may be missing a detail, but the GRCI method does not appear to require any assumptions regarding the noise distribution since they do not depend on the likelihood maximization approach. Hence, investigating the impact of a misspecification seems only relevant if the LSNM model is learned using an approach, such as in [9], that requires explicit assumptions about the noise distribution. Seeing this, it appears one does not need to worry about noise misspecifications if the GRCI approach is used.
> >
> > > Re: Questions, Scaling
> >
> > You mentioned that "in ANMs, scaling may lead to a flip in likelihoods". However, one would typically use ANMs for testing the independence between the residual and input (e.g., after fitting a model in a least-squares manner). Such a test should remain scale-invariant. Even if we look at the likelihood of ANMs, if the ANM is correctly specified (like you assume for the noise of LSNMs), shouldn't this also prevent flipped edges?
> >
> > > Re: Questions, Isn't Gaussian Distribution Merely a Convention
> >
> > Thank you for the clarification. I understand that g(X) is a scaling factor in the LSNM, but I was referring to the CAREFL model, which is characterized as X = f(X_pa, g(N)), where g is an invertible function. I apologize for the confusion due to overlapping notation. In CAREFL, there's no need to specify the correct noise distribution, since it aims to learn a model up to a (non-linear) invertible transformation of the noise which is sufficient. This is discussed, for instance, in Section 5 of "Counterfactual (Non-)identifiability of Learned SCMs" by Nasr-Esfahany et al.
> >
> > That being said, I understand that these points are generally less related to your work since you are rather investigating the impact of misspecifications in case of a likelihood approach to infer LSNMs.

---

> > > ### Author Response · Authors · 2023-08-12
> > >
> > > Thank you very much for the follow-up and clarification!
> > >
> > > > Re: Questions, Isn't Gaussian Distribution Merely a Convention
> > >
> > > CAREFL [11] employs $N$ not $g(N)$ as the model prior. Please refer to the following excerpts from [11]:
> > > - $u$ in Equations 5 and $z_j$ in Equation 6
> > > - "The prior of each flow is fixed to a Laplace distribution" - Figure 2
> > > - "All flows use an isotropic Laplace distribution as a prior." - Appendix F.1 Architectures and hyperparameters
> > >
> > > Therefore, CAREFL is in the form $Y=f(X) + g(X) \cdot N$, not $Y=f(X) + g_{1}(X) \cdot g_{2}(N)$ or $Y=f(X, g(N))$. So, learning an invertible transformation of the noise does not apply to CAREFL.
> > >
> > > Furthermore, if CAREFL were in the latter two forms, it would not be a LSNM. Given we are not aware of any identifiable SCMs in the latter two forms, the problem would also become ill-defined if CAREFL were in the latter two forms.
> > >
> > > > Re: Weakness, Little Discussion of GRCI
> > >
> > > The key theoretical insights from our work is that:
> > > 1. One **does** need to worry about noise misspecifications if a likelihood-based method is used for LSNMs, such as CAREFL-M [11] or LOCI-M [9].
> > > 2. One **does not** need to worry about noise misspecifications if an independence-based method is used for LSNMs. This can be any appropriate independence-based methods for LSNMs, such as the GRCI approach as you say or the proposed CAREFL-H.
> > >
> > > We provide theoretical analysis on why likelihood-based methods fail, to promote (any) independence-based methods as a solution for LSNMs in these difficult settings, including GRCI and the proposed CAREFL-H.
> > >
> > > > Re: Questions, Scaling
> > >
> > > Utilizing independence testing after a likelihood-based method to achieve high accuracy does not imply the likelihood-based method itself does not give flipped likelihoods (low accuracy). For example, on the right side of Table 1 in our paper, CAREFL-H gives high accuracy but CAREFL-M gives flipped likelihood (low accuracy). Please recall that the proposed CAREFL-H method is a combination of the likelihood-based CAREFL-M and subsequent independence testing.
> > >
> > > The likelihoods of ANMs can flip under correctly specified noise prior, too. This is mentioned on lines 159-161 and summarized in Figure 1 in our paper. Our experiments further support this point in Appendix O (please refer to lines 567-571).
> > >
> > > We sincerely appreciate the time you've dedicated to the discussion. Thank you very much!

---

> > > > ### Comment · Reviewer_ttx5 · 2023-08-13
> > > >
> > > > Thank you for correcting my understanding of the function class that CAREFL covers, this was helpful! While the results are limited to the LSNM setting, both the theoretical and empirical contributions have become clearer, and I see their benefits for future work. I am willing to increase my score to 6.

---

> > > > > ### Author Response · Authors · 2023-08-14
> > > > >
> > > > > Thank you very much for the positive comments!

---

### Official Review · Reviewer_GXXz · 2023-07-03

**Soundness:** 3 good
**Presentation:** 3 good
**Contribution:** 3 good
**Rating:** 7
**Confidence:** 3

**Summary:**

This paper explores causal discovery using heteroscedastic location-scale noise functional models (LSNMs) (which generalizes the additive noise models) and compares maximum likelihood model selection to causal model selection through residual independence testing.

**Strengths:**

1. I think this paper is significant in that it provides empirical and theoretical evidence of using the IT method for causal discovery.
2. This paper is well-written. The contribution is very clearly written from the introduction, so little chances to miss the contribution of the paper.
3. The simulations are extensive.

**Weaknesses:**

1. I think the robustness of the IT method should be more theoretically advocated.
2. Some results (e.g., Proposition 5.1.) don’t have proofs, and I am not sure they are trivial.

**Questions:**

- Q1. What is the implication and usage of Lemma 4.1 in the paper?
- Q2. What is the definition of noise misspecification and misleading CV? Does it mean that the estimation model for the residual (noise) is wrongly specified?
- Q3. “(Identifiability) When the form of the noise distribution is correctly specified, the causal model always has a higher likelihood (Figure 2a).” can be proved by Theorem 4.2 or Corollary 4.3? If so, could you please provide easy and intuitive explanations?
- Q4. I think the proof for Proposition 5.1 should be provided.

**Limitations:**

I believe that more theoretical analysis is necessary to claim the robustness of IT. For instance, does Proposition 5.1 imply that the IT-based method is robust against noise misspecification and misleading cross-validation? If we assume regression suitability in the machine learning method, will the result change for ML?

---

> ### Author Rebuttal · Authors · 2023-08-07
>
> ### Re: Weakness, 1 & Limitations, Robustness of IT Methods
>
> Please refer to the first global response (Re: Robustness of Independence-Based Methods Under Noise Misspecification in LSNMs) for the answer regarding the robustness of IT methods.
>
> ### Re: Questions, Q1
>
> The implication and usage of Lemma 4.1 is 3-fold:
> 1. (Appendix F) It is used to prove Theorem 4.2.
> 2. (Lines 147-150) It indicates that the performance of likelihood-based methods for LSNMs remains unaffected by misspecifying the parameters of the model prior, such as the mean and variance.
> 3. (Lines 159-161) It demonstrates that the estimated direction using likelihood in LSNMs does not flip when scaling the variances (under correct noise misspecification), which is a significant distinction from ANMs.
>
> ### Re: Questions, Q2
>
> The definitions of noise misspecification and misleading CVs are provided on lines 42-44 of the paper
>
> Noise misspecification refers to the situation where the noise of the estimation model is incorrectly specified by the user.
>
> Misleading CVs indicates the conditional variance in the ground-truth data being higher in the ground-truth causal direction and lower in the ground-truth anti-causal direction.
>
> ### Re: Questions, Q3
>
> The reference to the identifiability theorems from previous work can be found on line 147. These theorems establish that if the noise distribution is accurately specified in the model prior, and under the assumption that the data is generated by LSNM, the causal direction always has a higher likelihood than the anti-causal direction (except in some rare cases). These identifiability theorems are the foundation of likelihood-based methods for LSNMs.
>
> ### Re: Question Q4, Proof of Proposition 5.1 & Weakness, 2
>
> We did not give an independent proof of Proposition 5.1, because we felt it is too close to the analogous results for ANMs in Appendix A of [14] to present as a novel contribution. However, in Section 5.2 of our paper we present the underlying idea:
> 1. (Line 183-187) Under suitability, the reconstructed noise $\hat{N}$ approaches the ground-truth noise $N$ .
> 2. (Line 188-189) By employing a consistent HSIC estimator, the estimated HSIC value $\widehat{\text{HSIC}}$ converges to the true HSIC value $\text{HSIC}$.
> 3. (Line 192) In the ground-truth LSNM, $\text{HSIC}(\text{Cause}, \hat{N}\_{\text{Effect}}) = 0$ and $\text{HSIC}(\text{Effect}, \hat{N}\_{\text{Cause}}) \neq 0$ (i.e. identifiability of LSNMs).
>
> Together, we have $\widehat{\text{HSIC}}(\text{Cause}, \hat{N}\_{\text{Effect}})=0$ and $\widehat{\text{HSIC}}(\text{Effect}, \hat{N}\_{\text{Cause}}) \neq 0$ for LSNMs.
>
> ### Re: Limitation, Suitability for Maximum-Likelihood Methods
>
> Please refer to the first global response (Re: Robustness of Independence-Based Methods Under Noise Misspecification in LSNMs) for the robustness of IT methods under noise misspecification and misleading CVs.
>
> The suitability of regression methods is a valuable property for independence-based methods since they facilitate the reconstruction of the ground-truth noise, aiding in the determination of the causal direction using independence testing. However, the suitability of regression methods does not necessarily guarantee a higher likelihood for the causal direction compared to the anti-causal direction. Therefore, it does not offer the same value for maximum-likelihood based methods.

---

> > ### Comment · Reviewer_GXXz · 2023-08-13
> > **Response to the Rebuttal**
> >
> > Thank you for addressing my concerns and questions. I raise my score from 6 to 7.

---

> > > ### Author Response · Authors · 2023-08-14
> > >
> > > Thank you very much for reviewing our work and for your positive feedback!

---

### Official Review · Reviewer_bEFo · 2023-07-05

**Soundness:** 3 good
**Presentation:** 2 fair
**Contribution:** 2 fair
**Rating:** 3
**Confidence:** 4

**Summary:**

The authors investigate bi-variate causal inference with non iid, heteroscedastic noise. They specifically investigate a group of causal inference methods based on max-likelihood and present two main drawbacks: max-likelihood approaches with parametric assumptions on the noise distribution, such as normality, suffer when these distributions are misspecified. Further, the authors show that max-likelihood is inversely proportional to conditional variance. Hence, if the conditional variances are misleading the max-likelihood points in the anti-causal direction, which is the case for many pairs in the Tübingen real world cause-effect benchmark. The authors propose an independence test based approach fitting an affine flow to model the conditional density and HSIC test the residuals.


**Strengths:**

- Investigating the assumptions and failure modes of bi-variate causal inference methods is an important to ensure that these methods are accurate under real-world conditions.
- The authors provide a thorough argument against max-likelihood based methods based on theory and empirical evidence. The empirical experiments demonstrate how misspecified noise and noise level affect the accuracy of max-likelihood methods.
- The linkage of max-likelihoods failure modes to misleading conditional variance is plausible and is backed by theory.
- The proposed method CAREFL-H performs well in the benchmarks covering synthetic and real world data. It outperforms both independence test and maximum likelihood based approaches alike.

**Weaknesses:**

- The contribution of this work is limited in light of recent work. It partially overlap with the work on noise misspecification by Strobl & Lasko 2022, and the critic on likelihood-based causal inference by Schultheiss & Bühlmann, 2023.
- The pitfalls of independence test methods is not discussed, leading to an incomplete picture in regards to comparing both families of approaches.
- The proposed amendment to the CAREFL method introduces only an independence test afterwards and applies the same scheme (RESIT, GRCI) with different function modelling tools.
- The claim, that affine flows are "empirically suitable under noise misspecification and misleading CVs" is not sufficiently supported by the experiments which cover only limited function classes (sine, sigmoids).
- For instance, CAREFL-H performs worse than RESIT on SIM, which also uses HSIC with a different function estimator that works than the affine flows.
- Finally, the presentation including writing and figures is unclear at times (for details see minor remarks).

References
Schultheiss, C., & Bühlmann, P. (2023). On the pitfalls of Gaussian likelihood scoring for causal discovery. Journal of Causal Inference, 11(1), 20220068.
Strobl, E. V., & Lasko, T. A. (2022). Identifying patient-specific root causes with the heteroscedastic noise model. arXiv preprint arXiv:2205.13085.

Minor Remarks:
- Figure 2 plots are squished.
- Table 2a) is very cluttered and lacks a caption, maybe better as line chart to see trend.
- Theorem 4.2 is a main result of the paper but not explained sufficiently, i.e. only half a line long and the following paragraph is also very sparse.
- The acronyms ML and IT are associated with machine learning and information technology and hence easy to confuse, consider using different acronyms.

**Questions:**

- Table 1: If conditional variance is misspecified, i.e. higher in the causal direction, how do the M methods still get 100% right for all gaussian high noise pairs? And why does it not work for uniform noise?
- What could be some benefits of using max likelihood, for example, if learned function \hat{f} or scaling \hat{g} is misspecified, my intuition is that max likelihood has more slack than independence tests.
- What is the sample complexity of CAREFL-H, how much is it increased as both a train/test set is needed, where the test set needs to be large enough for an HSIC test with sufficient statistical power.
- Experiments: you test noise independence after a regression with CAREFL-H same as RESIT, why is RESIT better on SIM across the board?




**Limitations:**

The downsides of independence testing approaches are not addressed. Hence, the comparison between independence testing and max-likelihood methods is not complete. For example, in the homoscedastic case max-likelihood methods (CAM) outperform independence test methods (RESIT). This hints towards max-likelihood methods also having an edge against independence tests in some cases.

---

> ### Author Rebuttal · Authors · 2023-08-07
>
> ### Re: Contribution of the Work & Distinction to Previous Works.
> Please refer to the second global response.
>
> ### Re: Pitfalls / Downsides of Independence Test
> Our answer depends on the pitfalls you have in mind. If your concern relates to the sample inefficiency of independence-based methods, we addressed this matter in Section 6.1.2, line 255-260, as well as in our future work discussion on line 296. If your concern relates to the computational cost of independence testing, it is not a concern in our setting since we are targeting bivariate settings as mentioned on lines 2, 34 and 108.
>
> ### Re: Suitability
> Suitabilty basically means that an LSNM regression model has the capacity to predict the correct Y-value given input X (in expectation). It is well-known that since neural network models are universal function approximators, we can expect them to fit many regression functions, although it is difficult to prove suitability theoretically (line 203). Indeed our experiments show that suitability requires choosing hyperparameters to ensure sufficient model capacity (Section 6.1.2, lines 253 and 262)).
>
> For these reasons, we do not claim a priori that flow models are suitable (see lines 202-205), but we have provided empirical evidence that with careful training and hyperparameter selection, we can expect them to be suitable in practice. Table 2 summarizes our main experiments, including several non-linear functions (tanh, cosine, sine, and sigmoid) as well as difference ground-truth noise distribution (continuous Bernoulli, uniform, and exponential).
>
> ### Re: Questions 1.
> The answer to the question is directly provided by lines 147-150 and Theorem C.2.
> - Identifiability theorems of LSNMs establish that the causal direction consistently has a higher likelihood when the model noise prior is correctly specified. This conclusion remains unaffected by whether the conditional variance in the data is higher in the causal direction or not.
> - Regarding your example comparing Gaussian noise to uniform noise, as explained in the paper (e.g. lines 42-43, 127-134, and Table 1), the discrepancy in accuracy arises from noise misspecification, not the noise distribution itself. For example, specifying a uniform model prior when the true noise is uniform also yields high accuracy, which is proved by Theorem C.2.
>
> ### Re: Question 2. Benefits of using maximum likelihood.
> While our primary focus is on addressing the problem of noise misspecification and misleading CVs in LSNMs, where independence-based methods demonstrate superior performance, we have mentioned in the paper a few cases where likelihood-based methods may perform better:
>
> 1. Lines 261-262, when neural networks have low capacity to fit the functions $f$ and $g$, results the learned functions $\hat{f}$ or $\hat{g}$ to be misspecified. In this case, independence-based methods cannot utilize a good reconstruction of the noise.
> 2. Lines 155-156, when the noise distribution is known, the performance of likelihood-based methods is guaranteed by identifiability theorems.
> 3. Lines 255-260, with a small sample size and correctly specified noise distribution, likelihood-based methods may be more accurate. Independence-based methods require a larger number of samples to achieve accurate results. Intuitively, when the user managed to correctly specify the model, the likelihood-based methods can take advantage of this knowledge to be more sample efficient.
>
> ### Re: Question 3. Sample Complexity of CAREFL-H.
> The specific answer to this question depends on the architecture of the neural networks employed in CAREFL-H, but it is challenging to provide exact answer due to the lack of theoretical results for neural networks. However, we mentioned on lines 255 that CAREFL-M can have greater sample efficiency compared to CAREFL-H. For sample complexity of different HSIC estimators, please refer to relevant papers such as [5] and [14].
>
> ### Re: CAREFL-H VS. RESIT on SIM
> Did you mistakenly look at CAREFL-M instead of CAREFL-H? As indicated in Table 3(a), CAREFL-H demonstrates superior performance over RESIT on SIM benchmark in 3 out of 4 settings.
>
> ### Re: Limitations
> We have addressed the downsides of independence testing and cases when likelihood-based methods have an edge against independence-based methods in several places of the paper, as stated in earlier responses.
>
> For your example of CAM outperforming RESIT, it happens in the homoscedasitc settings of ANMs, which have been extensively discussed, and are not the main focus of our paper. Our paper primarily focuses on heteroscedastic settings of LSNMs, specifically under noise misspecification and misleading CVs. In these contexts, independence-based methods demonstrate superior performance.
>
> ### Re: Theorem 4.2 is Short
> Please note that some of the informal explanation and motivation for Theorem 4.2 and Lemma 4.1 appears **before** the mathematical statements since they flow from our empirical results in Table 1. All terms in the mathematical expression in Theorem 4.2 are defined. As mentioned on line 142, Theorem 4.2 demonstrates that the likelihood and conditional variance are negatively related. We utilize Theorem 4.2 as a foundation to derive Corollary 4.3 and the three relationships discussed on lines 153-158. In essence, the key insights and highlights of Section 4 (lines 142-158) stem from Theorem 4.2.
>
> ### Re: Acronyms
> We would like to clarify that we have explicitly defined these acronyms, ML (Maximum Likelihood) and IT (Independence Testing), in multiple places throughout the paper (lines 37, 78, 289, and 292) to ensure good readability and comprehension. "Machine Learning" or "Information Technology" terminologies are rarely mentioned in the paper, if at all. Therefore, we believe that the acronyms do not lead to confusion for the readers.
>
> ### Re: Captions
> We want to clarify that all (sub)tables and figures, including Table 2(a), do have captions.

---

> > ### Comment · Reviewer_bEFo · 2023-08-14
> >
> > Thank you for your response.
> >
> > Suitability is a very strong assumption that does not follow from the universal approximation theorem since there is no guarantee what function is actually fitted. Additionally, I again repeat the point that testing sine, sigmoid and tanh do not sufficiently support the claim "T is empirically suitable under noise misspecification and misleading CVs".
> >
> > This ties into the main criticism: the authors briefly mention that misspecifying $f$ and $g$ can hurt IT methods, i.e. lack of suitability. The title of the paper suggests a review of up and downsides of both classes, hence, we at least expect empirical simulations on the effect of function misspecification on IT and ML methods, as done for noise misspecification. Overall, even though sentences on this topic are found scattered across the paper, an actual investigation into the pitfalls of IT is missing. The analysis of the pitfalls of IT methods (such as sample complexity) in the rebuttal is appreciated and I most strongly suggest dedicating an entire section of the manuscript that discusses this, including theory and simulations.

---

> > > ### Author Response · Authors · 2023-08-14
> > >
> > > Thank you for the follow-up.
> > >
> > > ### Re: Guarantee of Suitability
> > >
> > > As mentioned earlier in our previous response and within the paper:
> > > - Theoretically, **it is difficult, if not impossible, to guarantee** suitability with neural networks (lines 202-203 in the paper).
> > > - Empirically, the universal approximation of neural networks indicates **the capacity of the neural network regressors being suitable** under careful fine-tuning and hyperparameter selection.
> > >   - Our experiments in Table 2 provide further empirical support for this, including several non-linear functions (tanh, cosine, sine, and sigmoid) and different noise distribution (continuous Bernoulli, uniform, and exponential).
> > > - Please also refer to the first global response (Re: Robustness of Independence-Based Methods Under Noise Misspecification in LSNMs)
> > >   - It explains why independence-based methods outperform likelihood-based methods under noise misspecification in LSNMs.
> > >   - The robustness of independence-based methods is further supported empirically by our experiments with 580 synthetic and 99 real-world datasets in Section 6.
> > >
> > > ### Re: Function Misspecification (the Reviewer's Main Criticism)
> > >
> > > - Misspecifying the functions $f$ and $g$ can hurt not only independence-based methods but also likelihood-based methods.
> > >   - In fact, nearly all machine learning tasks can suffer in performance due to different levels of function misspecification.
> > >   - Similar to numerous other studies, the ground-truth functions are unknown and learned by neural networks in our work.
> > > - According to the first global response (Re: Robustness of Independence-Based Methods Under Noise Misspecification in LSNMs), **misspecifying the functions $f$ and $g$ does not affect the robustness of independence-based methods as long as the conditional means and standard deviations are approximated**.
> > > - As stated in multiple places in the paper (e.g., lines 8-12, 41-49 & 58-59), we focus on investigating **noise misspecification** in LSNMs. Investigation on function misspecification (including empirical simulations on the effect of function misspecification as you say) is out of the scope of this work.
> > >
> > > ###  Re: Up and Downsides of Both Types of Methods & Pitfalls of Independence-Based Methods (Such as Sample Complexity)
> > >
> > > Referring to a previous response (Re: Question 2. Benefits of using maximum likelihood), the cases where likelihood-based methods outperform independence-based methods are already stated in the paper (lines 155-156 and 255-262). Furthermore,
> > > - It is important to note that **our work is not to conduct an exhaustive survey** comparing independence-based and likelihood-based methods in general. We reiterate that **our primary focus is under the settings of noise misspecification** in LSNMs (e.g., lines 8-12 in the abstract, 41-49 & 58-59 in the introduction). A broad-ranging comparison between these two method types in cases other than noise misspecification is out of the scope of this paper.
> > >
> > > To reiterate, if you refer to "pitfalls" as sample complexity or computational cost, we have addressed them in the paper and in the earlier response (Re: Pitfalls / Downsides of Independence Test).
> > >
> > > Last but not least, we do not consider sample complexity (or other "pitfalls" you may have in mind) of independence testing to be a substantial criticism against independence-based methods under the settings of interest, because:
> > > - We have extensively demonstrated in Table 1 and Section 6 that independence-based methods, **despite their potential for lower sample efficiency or other "pitfalls," outperform** likelihood-based methods across a range of hyperparameter choices and datasets. Likelihood-based methods **fail notably under noise misspecification, even without the "pitfalls"**.

---

> > > > ### Comment · Reviewer_bEFo · 2023-08-18
> > > > **Location-Invariant Noise in Synthetic Data Experiments?**
> > > >
> > > > Could the authors please comment on how the data generating process for the experiments in Sec. 6.1 are described in Appendix L, as well as in what way these are conducted on synthetic data with location scaled noise models?
> > > >
> > > > Unlike what you write ("ground truth LSNMs") it seems you actually evaluate on *non-location scale noise datasets* by Tagasovska et al. rather than the LS variants? Furthermore, I would disagree with calling the Tübingen cause-effect benchmark dataset 'synthetic' and while I immediately believe that some of its entries may contain location-scaled noise, most certainly not all.

---

> > > > > ### Author Response · Authors · 2023-08-18
> > > > >
> > > > > ### Re: Data Generation Process in Section 6.1
> > > > >
> > > > > As stated in the first sentence of Section 6.1 (line 225-226), the data generation process is provided in Appendix L (lines 539 - 545), including:
> > > > > - What LSNM SCMs are used (Equation 10):
> > > > >   - $Y := \tanh(X \cdot \theta_1 ) \cdot \theta_2 + e^{\cos(X \cdot \psi_1) \cdot \psi_2 } \cdot Z_Y$
> > > > >   - $Y := \sin(X \cdot \theta_1 ) \cdot \theta_2 + (\tanh(X \cdot \psi) + \phi) \cdot Z_Y$
> > > > >   - $Y := \sigma(X \cdot \theta_1 ) \cdot \theta_2 + \sigma(X \cdot \psi_1) \cdot \psi_2 \cdot Z_Y$
> > > > > - What noise distributions are used: continuous Bernoulli, uniform, exponential, beta, Gaussian and Laplace.
> > > > > - How parameters (e.g. $\theta, \psi, \phi$) are sampled following previous works.
> > > > >
> > > > > According to Equation 1 of the paper, these SCMs from Equation 10 are LSNMs, or the LS variants as you say.
> > > > >
> > > > >
> > > > > ### Re: Ground-Truth LSNMs are Non-LSNMs
> > > > >
> > > > > The datasets in Section 6.1 are generated by LSNMs and the datasets in Section 6.2 and 6.3 are not. We only used the phrase "ground-truth LSNM" in Section 6.1.
> > > > >
> > > > > The reasons to include these non-LSNMs in our evaluation are as follows:
> > > > > - In Section 6.2 and 6.3, we use the SIM benchmark and the real-world Tübingen cause-effect benchmark, respectively, for evaluation, because numerous previous studies in the domain of LSNMs utilize these two benchmarks for evaluation, e.g. [9, 29, 11, 30, 28].
> > > > > - Results from both LSNMs and non-LSNMs demonstrate that the proposed method performs well not only when the ground-truth SCMs are strictly LSNMs (i.e. Section 6.1) but also when the ground-truth SCMs are unknown or not strictly LSNMs (i.e. Section 6.3 & 6.2).
> > > > >
> > > > >
> > > > > ### Re: Calling the Tübingen Cause-Effect Benchmark "Synthetic"
> > > > >
> > > > > We have consistently referred to the Tübingen cause-effect benchmark as **"real-world"** throughout the paper (please see lines 52-53, 276). There is no instance in the paper where we refer it as "synthetic".

---

### Official Review · Reviewer_yfAc · 2023-07-10

**Soundness:** 4 excellent
**Presentation:** 4 excellent
**Contribution:** 3 good
**Rating:** 8
**Confidence:** 4

**Summary:**

This paper compares using maximum likelihood vs using tests of independence with regression residuals to select between a causal model and an anti-causal model, in order to infer the causal direction between two random variables. An empirical observation is that assuming a location-scale noise model, the maximum likelihood approach tends to perform poorly when the noise distribution is misspecified and the conditional variance in the causal direction is greater than that in the anti-causal direction. By contrast, the independence testing approach or at least the particular affine flow model used in this paper appears to maintain good performance even in such unfavourable circumstances. In addition to extensive empirical results to establish this phenomenon, this paper also provides some theoretical explanations of the phenomenon.

**Strengths:**

This paper is very well written and readable. The empirical results are convincing and illuminating. The theoretical results are interesting and provide a compelling and useful explanation of the empirical observation. Given the importance and difficulty of the problem of inferring causal asymmetry from observational data, the empirical results and theoretical analysis presented in this paper strike me as sufficiently significant.

**Weaknesses:**

As far as I am concerned, this paper does not have notable weaknesses. I would appreciate some explanation or even speculation on why the flow estimator remains empirically suitable under noise misspecification and misleading conditional variances. Does noise specification play any role in the flow estimator?

**Questions:**

Does the specification of noise distribution play any role in the flow-based independence tests? If not, I wonder if that is the main reason why it is more robust than the maximum likelihood methods. How much, if at all, do misleading CVs by themselves (i.e., without noise misspecification) affect the performance of the ML methods?

**Limitations:**

OK.

---

> ### Author Rebuttal · Authors · 2023-08-07
>
> ### Re: Weakness. Questions. Why Flows Remain Suitable Under Noise Misspecification. Role of Noise Specification.
> Please refer to the first global response (*Re: Robustness of Independence-Based Methods Under Noise Misspecification in LSNMs*) for our answer to these comments.
>
> ### Re: Question, How Much do Misleading CVs Alone Affect the Performance of Maximum-Likelihood (ML) Methods
> This question highlights another contribution of the work, demonstrating the differential impact of misleading CVs on the performance of ML methods in the context of LSNMs and ANMs. The details can be found in Figure 1, line 159-161 and Appendix O (line 567-571).
>
> In summary,
> - In LSNMs with correctly specified noise priors (i.e. without noise misspecification), the ML model selection score decreases for the true causal structure as CVs become more misleading, but **does not flip**, as implied by the theoretical identifiability results.
> - In LSNMs, adding noise misspecification to misleading CVs pushes ML scores over the edge of a wrong decision.
> - In ANMs,  misleading CVs **lead towards a wrong causal structure** even with correctly specified noise priors.
>
> This illustrates a difference between LSNMs and ANMs regarding their sensitivity to misleading CVs.

---

> > ### Comment · Reviewer_yfAc · 2023-08-14
> > **Helpful rebuttal**
> >
> > Thanks for the further, helpful clarifications. I am happy to keep my original score.

---

> > > ### Author Response · Authors · 2023-08-14
> > >
> > > Thank you so much for reviewing our work and the positive comments!

---

### Author Rebuttal · Authors · 2023-08-07

# Global Response to All Reviewers

Thank you very much for taking your time reviewing our work! We would like to answer some common questions here. We are sorry for the inconvenience.

Please refer to the reference section of the paper for the citation numbers from the response.

### Re: Robustness of Independence-Based Methods Under Noise Misspecification in LSNMs

Formal theoretical results for CAREFL-H are challenging to provide due to the involvement of neural networks in the flow estimators (lines 202-203). Informally, our intuition is that
independence-based methods are based on the **posterior** latent variable distribution $P(Z|X,Y)$ not the prior distribution $P(Z)$. A suitable regression method essentially learns to predict the correct $\hat{Y}$ value from $X$ (in expectation), and given $X$, $\hat{Y}$ implies a definite value for $\hat{Z}$, so the posterior is sharply peaked around the true value. We give a brief formal discussion to make these intuitions more precise.

In the context of the LSNMs, the ground-truth is $Y=f(X) + g(X) \cdot Z$ according to Equation 1. Let $\hat{f}\_{\theta}(X)$ be the conditional mean estimator, i.e. $\hat{f}\_{\theta}(X) = E[Y|X]$, and let $\hat{g}\_{\phi}(X)$ be the conditional standard deviation estimator, i.e. $\hat{g}\_{\phi}(X) = \text{STD}[Y|X]$. Here, $\theta$ and $\phi$ represent the parameterizations of the neural networks and may be omitted for clarity. We have:
- $\hat{f}(X) = E[Y|X] = E[f(X) + g(X) \cdot Z | X] = f(X) + g(X) \cdot E[Z | X] = f(X) + g(X) \cdot E[Z] $
- $\hat{g}(X) = \text{STD}[Y|X] = \text{STD}[f(X) + g(X) \cdot Z | X] = |g(X)| \cdot \text{STD}[Z | X]  = g(X) \cdot \text{STD}[Z] $

So, $\hat{Z} = \frac{Y-\hat{f}(X)}{\hat{g}(X)} = \frac{f(X) + g(X) \cdot Z - f(X) - g(X) \cdot E[Z]}{g(X) \cdot \text{STD}[Z]} = \frac{Z - E[Z]}{\text{STD}[Z]}$, where $E[Z]$ and $\text{STD}[Z] > 0$ are constants. Therefore, $\hat{Z}$ and $Z$ are identical up to shift and scale. Since $X$ and $Z$ are independent, so $X$ and $\hat{Z}$ are also independent.

To compare independence-based methods and likelihood-based methods under noise misspecification:
- As you can see from the math above, the specification of the prior distribution is **not involved**, indicating that the same independence result can be achieved with or without noise misspecification.
- However, as the model prior distribution is **involved** in likelihood computation (Equation 2 in the paper), when the model prior distribution is misspecified, the estimated likelihood may be incorrect.

This is the intuition why independence-based methods work better than likelihood-based methods under noise misspecification in LSNMs.

### Re: Contribution

Our main contribution is the study of **LSNM** models under **noise prior misspecification**, both through theoretical analysis and empirical evaluation (Line 55-59).  We provide comprehensive theoretical results for LSNMs under the challenging conditions of noise misspecification and misleading CVs. An extensive set of experiments evaluates the empirical performance of causal discovery methods in LSNMs. For all machine learning methods, it is important to understand how sensitive their performance is to their assumptions (such as knowing the noise prior distribution). Since the noise prior specifies a distribution over the unobserved factors, arguably it is especially difficult to specify it correctly based on domain knowledge.

While our proposed CAREFL-H algorithm is a member of the RESIT family of algorithms based on independence testing, the particular combination of fitting a neural net flow model with subsequent independence testing has to our knowledge not been previously evaluated by LSNMs. This combination is especially robust according to both our theory and experiments.

We would like to further clarify that our contributions are distinct and do not overlap with Strobl & Lasko 2022 (GRCI) [28] or Schultheiss & Bühlmann 2023 [23] for the following reasons:

- Unlike [28], which does not address the noise misspecification or misleading CVs, we identify and thoroughly analyze these challenges. Therefore, our theoretical contributions are novel and distinct from [28].

- While [23] discusses ANMs (as we mentioned on lines 96-97), our work primarily focuses on LSNMs. As highlighted on lines 159-161, our theoretical results for LSNMs and ANMs significantly differ, as supported by Lemma 4.1 and Proposition H.1. These distinctions for ANMs are further validated through empirical evidence, as presented in Table 5, Appendix O and Figure 5 (most materials about ANMs are in the appendix, as ANMs are not the focus of our work). Therefore, our findings do not overlap with [23].

---

> ### Author Response · Authors · 2023-08-21
>
> Re: Contribution (Continued)
>
> We would like to provide further clarification regarding the difference between our work and [23]. As we state in the paper (lines 96-97) and in the earlier global response (Re: Contribution), [23] focuses on ANMs (see Equation1 of [23]) and we focus on LSNMs.
>
> In Section 4 of [23], the authors discuss the complication of fitting data generated by a ground-truth ANM model with an LSNM model. Since the backward model of an ANM often takes the form of an LSNM, fitting data generated by a ground-truth ANM with an LSNM model may result in an incorrect causal direction. It is important to note that [23] does not address scenarios in which the ground-truth model is an LSNM. Therefore, despite their brief discussion of using LSNMs as the fitting model, the ground-truth model in [23] remains consistently ANMs. In contrast, our ground-truth model is LSNMs.
>
> Therefore, to reiterate, the theoretical findings of [23] contribute to ANMs, whereas our theoretical results contribute to the more general LSNMs.

---

### Decision · Program_Chairs · 2023-09-21

**Decision:**

Accept (poster)

**Comment:**

This paper is concerned with causal discovery based on heteroscedastic location-scale noise functional models, which are generalizations of the additive noise models, comparing likelihood-based model selection to residual independence testing-based method for finding causal discovery. The basic idea of the paper is interesting, and the empirical results are interesting and convincing. The theoretical results provide useful explanations of the empirical observations.  At the same time,  as pointed out by one of the reviewers, it has much overlap with Immer et al. (2022) and Schultheiss (2022), and the presentation of the paper is to be improved.  I suggest the authors try to make the paper more self-contained and make the unique contribution of the paper clearer.